# Combining nitric oxide and calcium sensing for the detection of endothelial dysfunction

Valeriia D. Andreeva [1,6], Haley Ehlers [2,3,6], Aswin Krishna R. C.[1,6], Martin Presselt[4,5], Lenie J. van den Broek[2] & Sylvestre Bonnet [1✉]

Cardiovascular diseases are the leading cause of death worldwide and are not typically diagnosed until the disease has manifested. Endothelial dysfunction is an early, reversible precursor in the irreversible development of cardiovascular diseases and is characterized by a decrease in nitric oxide production. We believe that more reliable and reproducible methods are necessary for the detection of endothelial dysfunction. Both nitric oxide and calcium play important roles in the endothelial function. Here we review different types of molecular sensors used in biological settings. Next, we review the current nitric oxide and calcium sensors available. Finally, we review methods for using both sensors for the detection of endothelial dysfunction.

According to the World Health Organisation, cardiovascular disease is the number one cause of death worldwide[1] and the leading cardiovascular disease is atherosclerosis. Atherosclerosis is a complex chronic inflammatory disease that develops over time due to chronic endothelial injury. The exact cause of atherosclerosis is unknown, and it often is not diagnosed until the late stages of lipid accumulation leading to plaque formation[2]. Unfortunately, the number of methods available for the detection of plaques remains limited. One non-invasive method for diagnosis, the cardiac stress test, only detects severely narrowed vessels of greater than 70%[3]. Still, most existing methods can only detect advanced stages of the disease. To diagnose earlier stages of the disease, and to gain a better understanding of disease progression, early non-invasive methods, such as small molecules sensors, should be used.

Some non-invasive methods for evaluating endothelial dysfunction (ED) have been developed in clinical studies. These methods include flow-mediated dilation of the brachial artery, intra-coronary Doppler Wire, cardiac magnetic resonance imaging, positron emission tomography, or multi-detector computed tomography[4]. Other non-invasive methods for diagnosis of diseases involving, especially abnormal blood vessel dilatation, are based on the detection of a signalling molecule called nitric oxide (NO)[5] that is present in the exhaled breath[6-9]. A variety of reviews on sensing NO in the exhaled breath to detect asthma, especially early diagnosis of exacerbations, and chronic heart failure, have been published[7,9-16], including reviews of the most recent techniques employed to detect gaseous NO[7,17]. However, the current methods for diagnosis are not extremely accurate and little is known about the exact initial causes of ED and disease progression.

ED can be defined as the earliest reversible precursor disease stage in the irreversible development of atherosclerosis[18-20]. ED has been linked to other diseases like nonalcoholic fatty liver disease[21], diabetes[22] and recently, SAR-CoV-2 (Covid-19)[23,24]. Endothelial cells regulate blood vessel homoeostasis through their production of NO as a vasodilator[25,26]. The generation of NO starts with increasing $Ca^{2+}$ concentration in the endothelial cell cytosol. In several works, it has

[1]Leiden Institute of Chemistry, Leiden University, Leiden, The Netherlands. [2]Mimetas B.V., De limes 7, 2342 DH, Oegstgeest, The Netherlands. [3]Leiden Academic Centre for Drug Research, Leiden University, Leiden, The Netherlands. [4]Leibniz Institute of Photonic Technology (Leibniz-IPHT), Albert-Einstein-Str. 9, 07745 Jena, Germany. [5]Sciclus GmbH & Co. KG, Moritz-von-Rohr-Str. 1a, 07745 Jena, Germany. [6]These authors contributed equally: Valeriia D. Andreeva, Haley Ehlers, Aswin Krishna R. C. ✉email: bonnet@chem.leidenuniv.nl

been shown that in the case of ED $Ca^{2+}$ intracellular concentration still fluctuates[27] while NO releasing by the cells has stopped[28]. These two obstacles make $Ca^{2+}$ and NO possible targets for ED diagnostic tools through simultaneous sensing of both analytes. Unfortunately, in vivo imaging has multiple limitations such as biocompatibility, cell bilayer penetration by molecules, intracellular targeting, light penetration, and signal-to-noise ratio, so new molecular probes and sensing methods must be developed[29,30]. This review provides an overview of recent developments in the modelling of ED, in the sensing of NO and calcium cations in biologically relevant conditions, and it gives an outlook for future research.

## ED modelling

**In vitro or in vivo?** Since the exact causes and dynamics of ED are not known, in vivo and in vitro models have been developed to study disease progression. Both in vivo and in vitro models have different benefits and drawbacks (Fig. 1). Animal models have been developed to study atherosclerosis, but next to ethical issues involved with animal experiments, the disease can manifest itself in different ways in animals than in humans[31]. Therefore, animal models may be seen as more realistic for testing new diagnostic tools, but they also have limitations for studying human disease progression[32]. Next to animal models, in vitro methods may further help to determine disease mechanisms, estimate disease risk levels, and help establish viable treatment options[33]. In vitro modelling started with 2D cell monocultures, which is a cheap and simple way to determine how cells react to different factors. More complex 2D cultures also exist that allow for the co-culture of more than one cell type. Usually, modelling atherosclerosis and ED involves the co-culture of endothelial and smooth muscle cells. Some methods of indirect 2D cell culture are using conditioned medium, microcarriers and bilayers or Transwell/Boyden chambers[33]. However, 2D cell cultures do not replicate the in vivo *3D* environment and cannot replicate the complex structure, extracellular matrices and cell-to-cell interactions present in animal models or humans[34].

3D methods allow cells to form structures that do not involve animal experiments but are more representative of what is found in vivo[35]. Current methods for co-culturing endothelial cells and smooth muscle cells in a 3D environment include the use of a scaffold or a gel for anchoring or separating the different cells[33]. To replicate a blood vessel with physiological shear stress, bioreactors with a pump have been used to grow the cells in a tube-like fashion and model how the cells react under

flow[36,37]. Microfluidic devices have been developed to culture cells in 3D and under shear. These devices are often manufactured from PDMS or another plastic[34]. Some microfluidic devices allow for the growth of endothelial tubes in a high-throughput manner, allowing for the study of endothelial cell interactions with immune cells, like monocytes[38] or a co-culture of endothelial cells and podocytes to create a human glomerular filtration model[39].

**Limitations of in vitro models**. Although in vitro models of ED are indispensable for optimizing ED diagnostic tools, these models have several limitations that need to be considered. The origin of the cells used in these studies needs to be considered carefully: cells from porcine, mouse or rat have been used in vitro, and the disease mechanism may vary between species, as also seen in vivo. For instance, mouse models have been used to study the immune system, shedding light on antibody synthesis and T cells. However, due to differences in immune systems, most immunological research done on mice does not translate directly to humans[40]. Another consideration within a given species, is the cell type that is being used. Often human umbilical vein endothelial cells (HUVECs) are used in in vitro studies; however, these cells might not be the best for modelling an arterial disease since they are vein cells from the umbilical cord and have different expression of genes and respond differently to environmental cues compared to artery endothelial cells[41]. Primary cells from human patients are an excellent source of cells for modelling diseases. However, these cells have to be used at a low passage number to maintain physiological relevance and often there is high donor-to-donor variability[33]. Finally, the culture substrate or vessel is often ridged in 2D culture and does not mimic the environment found in vivo[34].

However, new in vitro models are continuously being developed to help better study these diseases. Zhang et al. developed tissue engineered blood vessels using primary human endothelial colony forming cells, human coronary smooth muscle cells and human neonatal dermal fibroblasts with the addition of human monocytes to model early stage atherosclerosis[42]. Menon et al. created a microfluidic stenosis model to look specifically at the leucocyte to endothelial interactions[43]. Junaid et al. used the MIMETAS OrganoPlate® to look at the metabolic response of blood vessels to TNFα, which contributes the vascular stiffness in atherosclerosis[44]. For all these studies, the microfluidic systems still had a hard time mimicking the in vivo environment in regard to shear stress and flow patterns. In vitro models struggle to fully

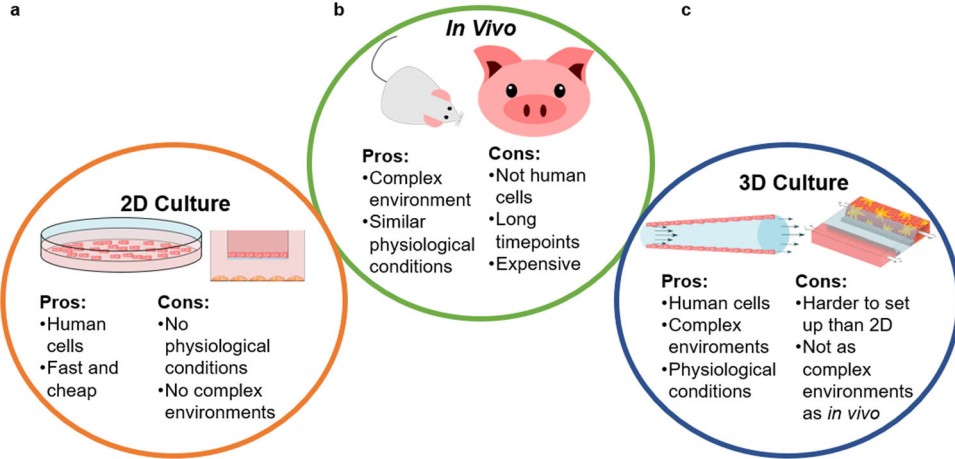

**Fig. 1 Pros and cons of in vivo and in vitro 2D and 3D cell cultures for modelling endothelial dysfunction. a** Pros and cons of 2D culture. **b** Pros and cons of in vivo culture. **c** Pros and cons of 3D culture.

recapitulate the dynamic in vivo environment and maintain the simplicity that can be consistent and reproducible. While great progress has been made in developing in vitro models for atherosclerosis and ED, each model still has its limitations, and a lot is still unknown about the disease's progression and prevention.

Recently, a lot of efforts have been made to develop molecular sensors that can be incorporated into cells for timely and spatially resolved monitoring of a selection of biomolecules in living cells. In this review, we further highlight NO and $Ca^{2+}$ sensors, used either in a cell-free context, in cells as an end-point assay or in living cells for constant monitoring of both analytes.

## Cell signalling pathways in the endothelium

**General considerations**. In a healthy artery, the endothelial cells and smooth muscle cells react to their changing environment, causing the artery to undergo vasoconstriction or vasodilation. The main radical involved in vasodilation is NO. NO has a relatively short half-life of approximately 6–30 s, so several biological processes lead to continuous NO production[45]. One of the factors leading to an increase in NO production is the change in cytosolic $Ca^{2+}$ inside the ECs. This change in intracellular $Ca^{2+}$ is due to influxes of extracellular $Ca^{2+}$ into the cytosol and the release of $Ca^{2+}$ from calcium ion reserves inside the endothelial cells[46]. The cytosolic $Ca^{2+}$ binds to calmodulin (CaM), which activates an enzyme called endothelial nitric oxide synthase (eNOS)[47]. eNOS is coupled to tetrahydrobiopterin ($BH_4$), which stabilizes the active dimeric form of the enzyme (Fig. 2a). When intracellular $Ca^{2+}$ leads to the activation of eNOS NO is produced by converting L-Arginine into L-Citrulline[48]. NO being a small neutral molecule it can diffuse through the cell membrane into the smooth muscle cells, causing them to relax and the blood vessel to dilate.

Several factors may lead to an increase in NO production, like shear stress, hormones, and proteins. For example, acetylcholine, bradykinin or histamine led to an increase in intracellular $Ca^{2+}$ leading to NO production. Shear stress, hormones and growth factors activate the eNOS enzyme through phosphorylation[47]. Fluid shear stress helps maintain a basal NO production leading to the maintenance of vascular tone[49]. With certain stimuli, like increased shear stress, the eNOS enzyme can be activated without a sustained increase in $Ca^{2+}$ [49]. While the regulation of eNOS can be controlled by several cascades of events, the pathophysiological changes associated with eNOS in regard to ED are still not completely understood. It is believed that there are many factors that lead to a decrease in NO production. Patients, with major cardiovascular risk factors like smoking, diabetes, hypertension, ageing or hypercholesterolaemia, have proven decreased NO production[28]. These risk factors lead to increased production of ROS in the vessel wall, in turn leading to a reduction in the bioactivity of NO by eNOS inactivation and a decrease in NO production through eNOS uncoupling[28]. $BH_4$ oxidation and depletions due to oxidative stress is a factor leading to eNOS uncoupling[50]. A decrease in NO production due to the uncoupling of eNOS is the leading characteristic of ED (Fig. 2b). Therefore, it is important to fully understand how NO is produced to study disease progression.

**Role of intracellular calcium and NO in endothelial and smooth muscle cells**. Vascular tone is directly related to the amount of blood vessel constriction relative to the maximal dilation. In diseased or atherosclerotic vessels, there is a decrease in vascular tone. Hyperpolarization of the endothelium and smooth muscle cells, through increases in intracellular $Ca^{2+}$, is one way to regulate vascular tone[25]. This hyperpolarization of the

cell membrane of the endothelium and smooth muscle cells has been termed endothelium-dependent hyperpolarization (EDH). NO is a known factor that can cause the smooth muscle cells to undergo EDH. However, the change in intracellular $Ca^{2+}$ can also lead to EDH without the synthesis of NO[25]. Félétou nicely summarized the role of intracellular $Ca^{2+}$ on EDH and ED in his review article[25].

In order to study the effects of NO and $Ca^{2+}$ in the endothelium and smooth muscle cells, understanding the dynamics of these molecules as a function of time is crucial. As previously discussed, NO production decreased in cells with ED. For example, Balbatum et al. used electrodes to trace NO in individual healthy and diseased single endothelial cells[51]. Figure 2c, d illustrates the decrease in NO produced by a healthy endothelial cell compared to a hypertensive cell after stimulation by calcium ionophore A23187. Not only did their study show a difference in maximum NO generated, but the NO decay rate was significantly different between the healthy and diseased endothelial cells[51]. The dynamics of the evolution of $Ca^{2+}$ concentration differs from that of the concentration of NO. Even unstimulated endothelial cells have changes in $Ca^{2+}$ level, typically referred to as $Ca^{2+}$ "puff" (*intra*-cellular) or $Ca^{2+}$ "waves" (*inter*-cellular)[52]. Burdyga et al. illustrated the changes in $Ca^{2+}$ concentration (hereafter, $[Ca^{2+}]$) using a calcium-dependent fluorophore and showed how the changes in $[Ca^{2+}]$ in a cell vary depending on location (Fig. 2e–g). They measured $[Ca^{2+}]$ in situ in blood vessels in the endothelium and smooth muscle cells. While calcium waves can propagate from one endothelial cell to another, in smooth muscle cells, the changes in $[Ca^{2+}]$ appeared as calcium "sparks" that did not spread from one cell to another but led to vasomotion[52]. The difference between calcium sparks and puffs depends on the family of calcium channel[53]. It has been shown that the $Ca^{2+}$ waves and oscillations in the endothelial cell regulate NO production which in turn regulates $Ca^{2+}$ sparks in smooth muscle cells, thus leading to changes in smooth muscle cells' force[25,26]. In order to fully understand the causes of ED and disease progression, it is important to be able to characterize and monitor in time and space the changes in NO and $Ca^{2+}$ concentrations within different cell populations of the blood vessel. Overall, we are still missing an integrated view of the time and space evolution of the concentration of $Ca^{2+}$ and NO and a global understanding of the role of their coupling in ED.

## Towards molecular probes for the sensing of ED

**Different types of sensors for bioanalytes**. As mentioned above, electrodes sensitive to molecular gases offer after calibration direct reading of gas concentrations in a medium, and this is in a time-dependent fashion. Such electrodes exist for several gases, notably for NO, CO, $O_2$, or $H_2$. On the other hand, only a few molecules can be probed with such electrodes, and most biological analytes (e.g., $Ca^{2+}$ ions) cannot. Spatial resolution and intracellular localization are also difficult to achieve for such electrodes, the tip of which remains big compared with cells. Extensive efforts have hence been made in the past decades to develop molecular probes for sensing, i.e., molecules that modulate their optical or magnetic properties in the presence of a particular biological analyte. Such sensor molecules are mostly used for intra-cellular sensing using optical microscopy; however, they can also be used to functionalize classical electronic devices to improve selectivity; NO-sensitive chemoresistors, for example, utilize the specific affinity of metalloporphyrins or corroles to NO for detecting NO in exhaled breath or gas mixtures[6,7]. In comparison with conventional electrodes, molecular probes offer the advantage of being tiny, easily operated, highly sensitive, and potentially very selective, which offers quick optical readouts with

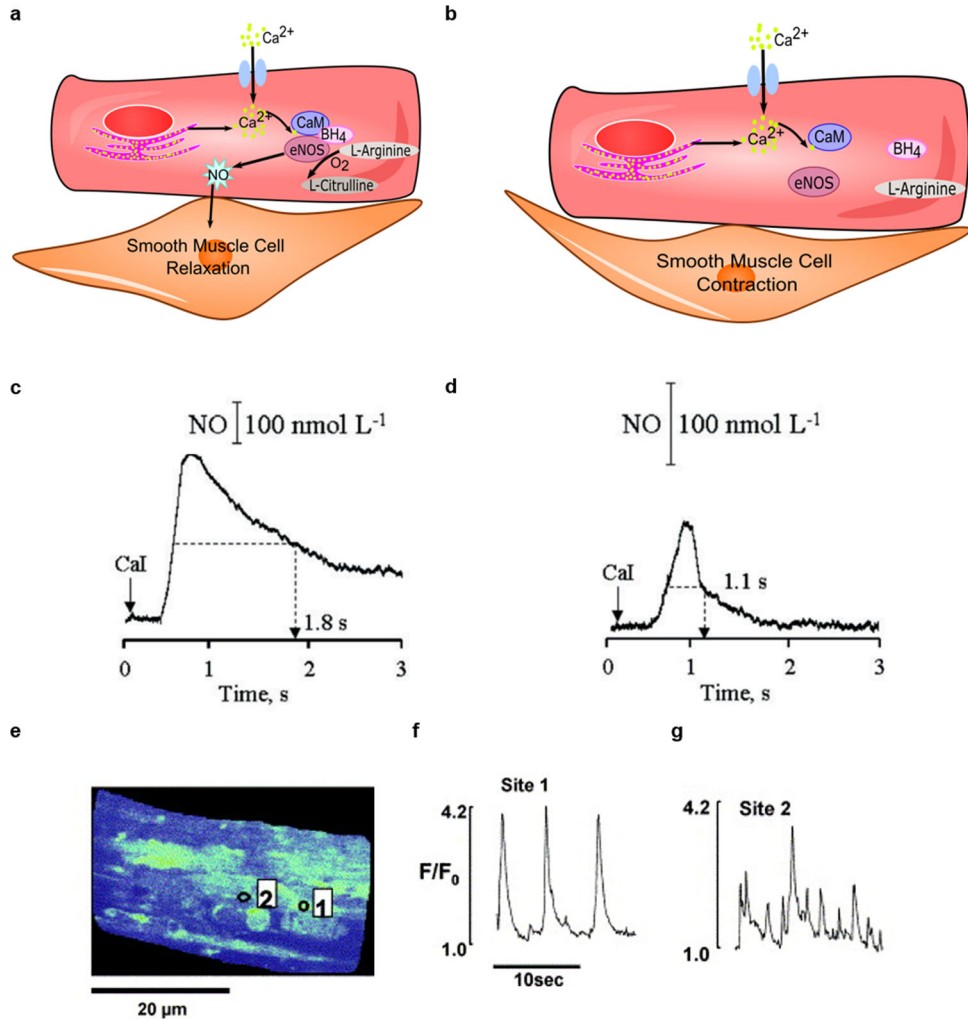

**Fig. 2 Nitric oxide and calcium in endothelial cells in healthy and disease setting. a** The cell signalling pathway from intracellular $Ca^{2+}$ to nitric oxide in homoeostatic endothelial cell, and **b** in a stressed endothelial cell. **c**, **d** NO release from single endothelial cell after stimulation with Calcium Ionophore A23187 (CaI) in vitro from iliac artery of **c** normotensive rat and **d** hypotensive rat[51] as measured with a single fibre porphyrinic sensor. **e–g** Calcium signalling pattern in endothelial cells under resting conditions (Reprinted with permission from ref. [51]). **e** X–Y image of single endothelial cells showing calcium oscillations in sites 1 and 2 (marked). **f** Calcium concentration vs. time at the site 1. **g** Calcium concentration vs. time at the site 2[52] (Reprinted with permission from Elsevier (Burdyga et al.[52], Copyright 2003)).

minimal invasiveness for the cells. In addition, they can measure the concentration of an analyte inside the cell, and they can be adapted for probing a wide range of bioanalytes, such as NO or $O_2$, but also radicals (superoxide, $HO^\bullet$, etc.), metal ions ($Ca^{2+}$, $Fe^{2+}$, etc.), proteins, etc. On the other hand, the large number of published molecular probes has made it increasingly difficult to select one for a particular application, and the marketed selectivity might be challenged in the complicated environment of a cell (e.g., NO probes might sense HNO or superoxide, etc.). Every probe has its advantages and disadvantages, which we shortly discuss below. Therefore, several critical properties should be considered for selecting molecular sensors for detecting ED. In this part, we discuss general properties such as solubility, working wavelength, brightness, toxicity, intracellular targeting, and photostability, and in parts "Nitric oxide sensors" and "Calcium chemosensors" we will focus on NO and $Ca^{2+}$ probes, respectively.

**Solubility, localization, encapsulation**. Small molecular probes are characterized by their solubility and hydrophilicity (log P). Both properties directly influence the cellular uptake and intracellular localization of the probe, which in turn influences how it may interact with its targeted analyte. For example, hydrophobic probes may target membranes, while water-soluble ones may end up in the cytosol. Another example is that positively charged hydrophobic cationic species might end up in the mitochondria, which have a negatively charged membrane, while negatively charged molecules may end up in the acidic environment of the lysosome. However, many details of the molecular formula of a probe matter, so general rules are difficult to be drawn. More specific intracellular targeting of fluorescent probes can be obtained by modifying the probe with a functional group that "tags" the probe for specific transport to the organelle of interest. For example, mitochondria targeting has been achieved by covalent modification with a triphenylphosphonium group or by quaternized pyridines[54,55], while morpholine is often used as a lysosome-targeting group[56].

Molecular probes that are in principle selective for a given analyte are also susceptible to interact with non-targeted biological substances also present in the complex cellular environment. In general, protecting the probe from undesired biological interactions, to control selectivity and localization, may

become very important. Such protection is typically performed either by encapsulation of these probes into polymeric nanoparticles or by assembling a PEG-based shell around the probe[57]. Also, the charge and the nature of the functional groups at the surface of these encapsulation molecules influence cell penetration property and intracellular localization.

**Working wavelength**. In principle, all visible wavelengths are eligible for intracellular fluorescent probes. However, fluorescent probes with both absorption and emission wavelengths in the spectral range of 600–900 nm[58], which describes the range comprising the red to the near-infrared region (NIR) of the spectrum, are of higher interest. They allow for low autofluorescence interferences, deeper tissue penetration properties (for in vivo imaging), and minimal photodamage to biological samples, which are critical in particular for long timescale observations. Therefore, NIR fluorescent probes are often considered the best choice when it comes to the detection of biologically relevant analytes such as $Ca^{2+}$, NO, superoxide, etc. On the other hand, many NIR dyes require extended polyaromatic structures, which may lead to high hydrophobicity, problematic aggregation, and off-toxicity. Also, NIR light is not visible to the naked eye or conventional cameras, thus NIR probes often require special techniques or equipment to visualize the signal. Conventional NIR emitters also suffer from large non-radiative decay rates and low quantum yields[59–61]. The narrow optical energy gap in the NIR region results in more non-radiative processes from the first excited state ($S_1$) to the ground state ($S_0$) to occur as stated by the energy gap law[62]. Newer strategies that consist in employing intermolecular charge-transfer and charge-transfer aggregates via nonadiabatic coupling suppression in organic emitters have been developed recently to overcome these energy-gap law related limitation of NIR emitters[63]. Finally, when looking at different analytes at the same time, it is critical to have several probes that can be excited and detected in different regions of the spectrum, allowing multicolour imaging[64,65]. Overall, this domain of research is very active, and many specific molecular probes have been developed based on blue, green, red, far-red, or NIR fluorophores[66,67].

**Brightness**. The luminescence brightness of a molecular probe is defined as the product of its molar absorption coefficient at the excitation wavelength ($\varepsilon$) and its luminescence quantum yield[68]. Since most excitation and emission light is lost due to light scattering and absorption inside the cellular environment, the brightness of a probe is a very important parameter. The higher the brightness, the higher the signal-to-noise ratio will be during imaging, therefore allowing deeper light penetration and deeper tissue imaging. Moreover, contrast is also critical for imaging, so the molecular probe should show large differences in brightness between the on and off states of the probe.

**Toxicity and phototoxicity**. One important characteristic of a sensor is that it should be non-toxic to cells. In vivo toxicity induced by probes may result in severe organ dysfunction and disease. In vitro, it may lead to cell death and destruction of the model of the endothelium. It is important to note that toxicity is often dose-dependent and species-dependent. Therefore, cell toxicity measurement should be carried out before going to in vitro testing, and systemic toxicity should be investigated before performing in vivo experiments. Luminescent probes need to be irradiated by light to perform imaging. Many molecules, upon light excitations, are capable of transferring the photon energy or an electron from their excited state to a nearby biological acceptor such as molecular oxygen, a process known as the "photodynamic" effect. Such mechanisms lead to the formation of reactive oxygen species, which increases oxidative stress and may lead to cell death during microscopic observation. The phototoxicity of molecular probes is also an important parameter that needs to be checked.

**Photostability**. Of course, the chemical stabilities of many molecular probes may be compromised after intracellular uptake, for example via the action of metabolic enzymes such as P450s or esterases. In addition, the photostability of a molecular probe is another important consideration when it comes to bioimaging applications. For high-resolution imaging, high light intensities are often required, which may lead to the photodestruction of the probe. This photodestruction is often a consequence of the photodynamic effect introduced above: the singlet oxygen that may be formed from the excited state of the probe is highly reactive and may oxidize the organic structure of the probe itself (instead of endogenous biological molecules), thus leading to a full loss of the emission properties of the exogenous probe. Photostability varies greatly with the probe. For example, probes with linear extended conjugation like cyanine dyes, are more prone to photobleaching than cyclic aromatic systems like porphyrins, aza-BODIPYs, etc.[68], while transition metal-based probes are often more photostable. Also, once these probes are taken up by lysosome, most fluorophores, including fluorescein, BODIPY, and cyanine derivatives, lose fluorescence within several days[69,70], which makes long timescale imaging a great challenge.

**Sensing mechanism**. Box Fig. 1a depicts regular light absorption and emission for a fluorescent dye. The molecule in the singlet ground state $S_0$ absorbs a photon, and one electron in the HOMO orbital jumps into a higher energy level (typically the LUMO), resulting in a singlet excited state $S_1$. From the LUMO, it will relax back to the HOMO either by emitting energy in the form of a photon, or by non-radiative decay. The PET quenching mechanism (Box Fig. 1b) differs from Box Fig. 1a at the relaxation stage of the $S_1$ excited state of the probe. Usually, the dye has an electron-rich functional group in its structure, called PET donor, which puts a high-energy filled molecular orbital (QO) in the immediate vicinity of the dye group. One electron of the PET donor group may be transferred, in the excited state $S_1$ of the dye, into the lowest-energy singly occupied orbital (SOMO) of the excited state. The electron in the resulting highest-energy SOMO orbital from the photo-reduced dye, relaxes back into QO, recovering the ground state $S_0$ of the sensor molecule. In this non-radiative decay process, the photon energy is lost into molecular vibrations, and the photon emission observed in the unquenched dye (a) does not take place: the emission is quenched by the PET donor. In many $Ca^{2+}$- or NO-sensing molecules, the PET quenching mechanism takes place in the unbound state of the probe, which usually contains high-energy, nitrogen-based electron donor groups in their structure (e.g., a tertiary amine in the BAPTA calcium chelator, or two primary amines in a phenylenediamine NO-binding unit). Therefore, the sensor's emission is low in the absence of an analyte.

Upon binding of the analyte to the chelate, PET quenching is prevented by the interaction between the calcium ion or NO molecule, and the electron-rich functional group of the probe. This interaction dramatically lowers the energy level of the QO orbital, thus preventing PET quenching to occur (Box Fig. 1c). Therefore, such a molecule is not quenched by PET in the presence of the analyte, and the probe has increased emission, compared to the analyte-free situation. As the amount of PET quenching and the emission quantum yield depends on the amount of analyte-bound sensor molecules, this mechanism can be utilized to determine the analyte concentration (see Box 1).

**Box 1 ▌ PET sensing**

*Photoinduced electron transfer*, or PET, is a mechanism where the light emission of a dye in its excited state is quenched by an electron donor, thus affording little or no emission[179]. When PET "quenches" the emission, the dye molecule absorbs a photon, but the resulting excited state returns to the ground state either without the emission of a visible photon, or with reduced emission intensity. The reduced emission intensity typically occurs because other photophysical processes, called non-radiative processes, are faster than photon emission. When such a quenching mechanism is cancelled, for example upon the interaction of the PET donor with a calcium ion or NO, the dye molecules recover their ability to emit a photon. The number of emitted photons, divided by the number of photons absorbed by a given number of dye molecules, is called the emission quantum yield. The difference between the energy of the emitted photon, and that corresponding to the absorption maximum of the dye, is called the Stoke shift; it is usually given in eV.

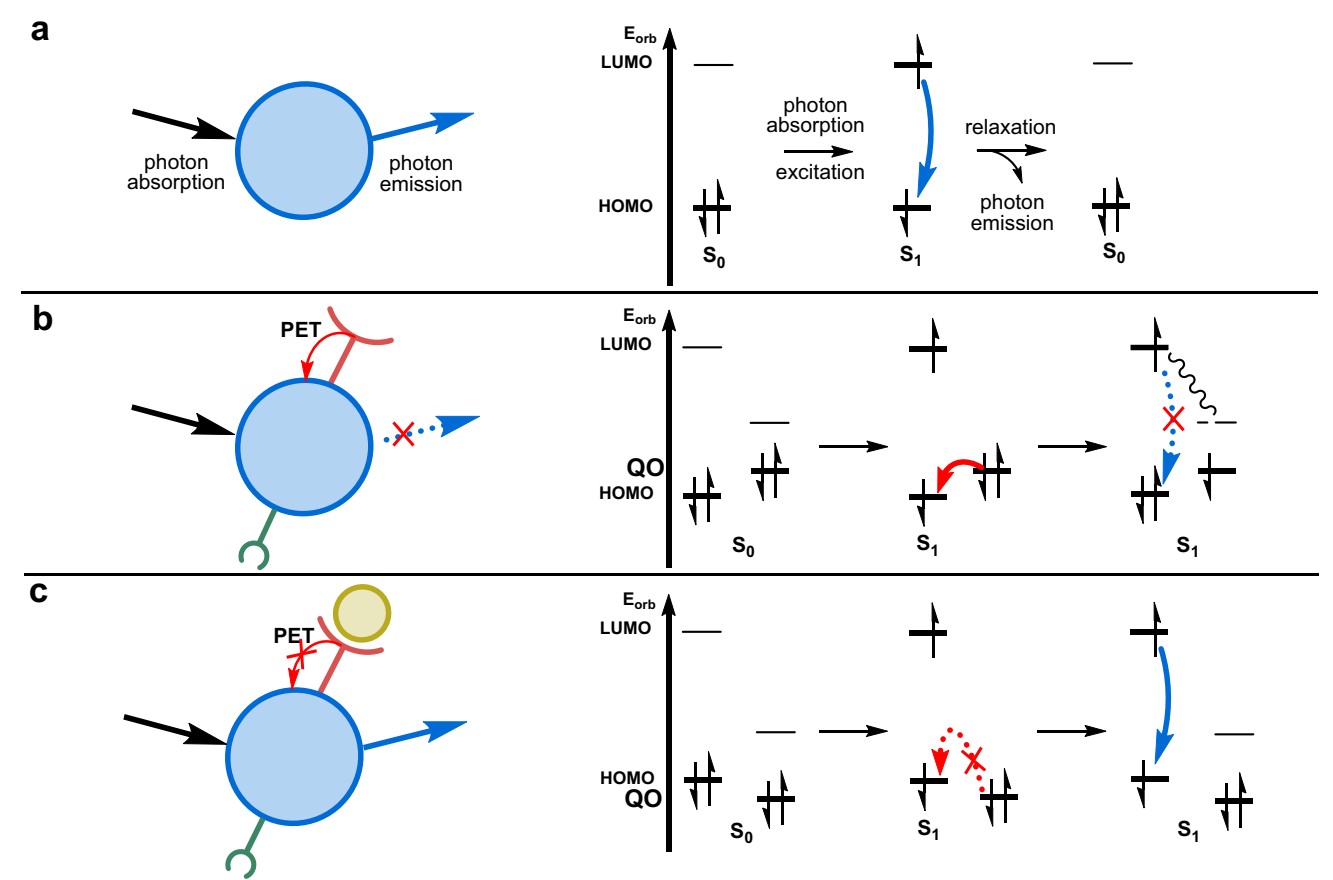

**Box Fig. 1 Mechanism of photoinduced electron transfer. a** Fluorescence emission in a fluorescent molecular probe (blue sphere). PET quenching of the fluorescence emission of a molecular probe for $Ca^{2+}$ ions **b** in the absence of $Ca^{2+}$ (red part is the calcium chelate) and **c** in the presence of $Ca^{2+}$ (brown sphere). PET photo-electron transfer.

## NO sensors

**Molecular properties of NO**. NO or nitrogen monoxide was identified as a secondary messenger molecule in the late twentieth century[71,72]. It was a great surprise for the scientific community to realize that a radical molecule like NO was of biological relevance, as radicals were considered to be toxic species. The repercussions and implications of this finding were later recognized with the Nobel Prize in Physiology and Medicine (1998), awarded jointly to Robert F. Furchgott, Louis J. Ignarro and Ferid Murad, "for their discoveries concerning NO as a signalling molecule in the cardio-vascular system"[73]. Chemically, NO is a diatomic free radical molecule that forms a gas at 1 atm and 25 °C. It has one unpaired electron in an antibonding singly occupied molecular orbital (SOMO). Because NO has a low dipole moment (0.159 D)[74], it forms relatively weak intermolecular interactions. In comparison to other radicals, NO radical is relatively stable, as it does not dimerize

easily. NO is also both hydrophobic (solubility in water is only 1.94 mmol L$^{-1}$ at 25 °C) and hydrophilic (its water–octanol partition coefficient log $P_{o/w}$ is 0.74)[75,76]. The diffusion of NO through cell membranes is hence very rapid, characterized by permeability coefficients from 18 to 73 cm s$^{-1}$ for lipid membranes[77–79]. Unlike other free radicals, such as superoxide $O_2^{•-}$ or the hydroxy radical OH$^{•}$, NO is a poor oxidant and a poor reducing agent under physiological conditions[80–83]. Additionally, NO reacts very specifically towards biological targets like oxyhemoglobin, methemoglobin, soluble guanylate cyclase (sGC), cytochrome c oxidase (CcOx), and more (Fig. 3h). All these properties facilitated the evolution of NO as a molecule in biology.

**NO sensing: general aspects**. Over the past two decades, many techniques have been developed to detect NO in tissue or exhaled

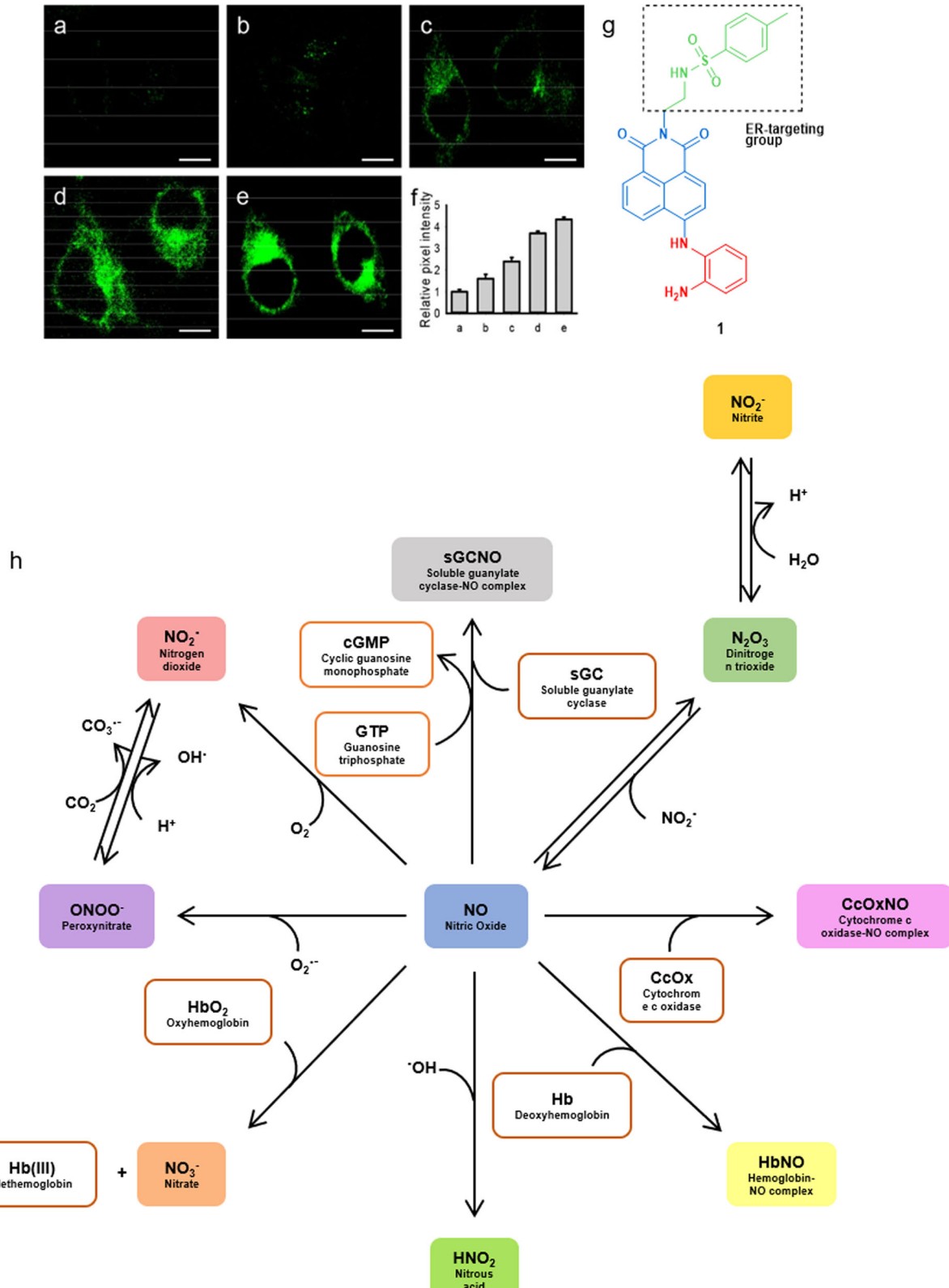

**Fig. 3 Targeted fluorescent imaging of endogenous NO and different pathways for biological consumption of NO.** Two-photon fluorescence images of HeLa cells treated by different concentrations of ER-stress inducer tunicamycin for 12 h, **a** 0, **b** 0.5, **c** 2.0, **d** 10.0, and **e** 100 μg mL$^{-1}$, and then stained by the NO-sensitive molecular probe **1** (10 μM) for 20 min. Scale bar is 10 μm (Reprinted with permission from ref. [94]. Copyright 2018 American Chemical Society). **f** Relative pixel intensities for images **a**–**e**. Tunicamycin is a nucleoside that is commonly used to induce endoplasmic reticulum stress. **g** Chemical structure of the fluorescent probe **1**. **h** Nitric oxide consumption in the biological environment.

breath[7,17]. Such techniques include direct detection of NO absorbance[7,17], electrochemistry[84–86], paramagnetism[87], chemiluminescence[88], and absorption or fluorescence changes upon NO-binding to molecular dyes[89,90]. NO-binding molecular dyes, in particular, can be used to functionalize (semi)conductors or transparent supports for external electronic or photometric NO-detection devices based on absorption changes. They can also be directly added into living systems to realize intra-cellular NO detection based on bioimaging of the local changes in optical emission of the dye due to its interaction with NO. In fact, bioimaging techniques have made life easier for biologists to carry out in vitro and in vivo studies on NO. It has always been difficult to design a sensing device suitable for direct, rapid, non-invasive, and timely and spatially resolved, detection of NO in living systems. The combination of bioimaging techniques with NO-sensitive molecular probes is, on the other hand, ideal for that purpose, because it enables in situ, in vitro and in vivo observations that are direct consequences of the presence of NO in tissues and cells[91]. Fluorescence imaging has numerous advantages over other NO imaging techniques such as chemiluminescence or electron paramagnetic resonance spectroscopy: it is extremely sensitive, selective, spatiotemporally resolved, and easy to realized experimentally, using optical microscopes that are readily available in most laboratories[92,93]. As the fluorescence signal and localization of molecular probes can be drastically modulated by chemical modifications, sensors relying on light of different power and wavelengths, and localizing in different regions of cells, can be prepared. NO-sensing fluorescent probes may even provide dynamic information concerning the localization and concentration of NO molecules. For example, Fig. 3a–f depicts an end-point assay with 1, a NO-sensitive molecular probe targeted to the endoplasmic reticulum (ER). It is capable of imaging the level of exogenous and endogenous NO in ER of a living cell during tunicamycin-induced ER stress[94]. In many of these studies, interdisciplinary collaboration between chemists, biologists and medical doctors, is necessary, to refine the properties of the molecular probes necessary for diagnostic applications.

**Challenges in NO sensing**. NO exerts its biological effects via the direct and reversible interaction with specific targets, for example, soluble guanylate cyclase (sGC) or through the generation of secondary reactive nitrogen species (RNS, Fig. 3h), many of which are also radical species. Because of this complexity, designing a chemical probe capable of selectively reacting with NO is challenging. Also, "measuring" NO usually means localizing NO in cells, as well as measuring its (relative) local concentration. The widely accepted physiological concentration of NO is in the range of 1–100 nM;[95] it has been determined mostly using NO-specific electrodes. This rather low value asks for highly sensitive molecular probes capable of reacting with minute amounts of NO. Moreover, NO is typically short-lived in a biological environment, with a half-life of 1–3 s[96,97]. This intrinsic instability also sets kinetic requirements for NO detection by molecular probes: basically, the probe should react quickly upon generation of its substrate, before NO degrades by reaction with other molecules. Of course, low toxicity, a suitable range of physiological pH, and selectivity concerning other interfering molecules are other properties that need to be considered when talking about molecular probes for NO sensing.

As explained in section "Limitations of in vitro models", NO diffuses out of endothelial cells to smooth muscle cells and blood cells after it is synthesized from L-arginine in the endothelial cell[98,99]. Therefore, an NO molecular probe can in principle remain outside the ECs, and does not need to be taken up in the cytosol of ECs or SMCs. However, the short half-life of NO in the

biological environment is also a big concern because if diffusion from the endothelium to SMCs or RBC is slow, then detection should take place before NO has disappeared. Moreover, for accurate measurement of the NO levels in diseased cells, kinetic and compartmentalization aspects of NO need to be considered. To understand these kinetic aspects, one would ideally need time-dependent sensing of NO at various cellular targets, which despite the massive amount of work realized in the field, has remained to date an unravelled challenge. Finally, even without considering kinetic aspects, one should realize that it is not trivial to quantify absolute NO concentrations in physiological environments using molecular probes, nor to unravel the influence of media heterogeneity on reaction kinetics.

**Existing NO sensors**. Over the past three decades, many sensors have been developed to analyse NO generation and distribution in living cells[100,101]. Among those, a few electrochemical sensors as well as a few organic probes, metal-based probes and nanoparticle-based probes employing fundamentally different NO sensing groups have been developed.

*Electrochemical NO sensors*. Electrochemical NO sensors are small electrodes that allow direct sensing of the NO level. They have been used both in vitro and in vivo. Enhanced sensitivity of on-site measurements, as well as rapid response (at least 100 ms), are among the major advantages of direct electrochemical measurements. The most commonly used material for the working electrode of these sensors includes platinum and its alloys, carbon fibre, and gold[102]. The electrode is coated with a permselective membrane capable of the electrooxidation or electroreduction of NO as well as a mechanism for discriminating electroactive interferences. During a measurement, a sufficiently positive or negative potential at an electrode surface is applied to electrochemically oxidize or reduce NO. The resulting current at the electrode surface is measured, which is proportional to the concentration of NO in the solution. A major disadvantage of this technique is the occurrence of biofouling (adhesion of platelets and blood proteins) during the measurements. This biofouling resists the usage of such sensors in the blood (i.e., protein adsorption, platelet adhesion, and thrombus formation) and tissue (i.e., fibrous encapsulation and infection). Even though strategies to reduce biofouling by passive protection of sensors using membranes like Nafion and polyurethanes have been proposed, biofouling still results in poor reproducibility and sub-optimal analytical performance[102]. Another disadvantage of these sensors is that the oxidation or reduction of many other electroactive molecules (e.g., glutathione, $H_2O_2$, $NO_2$, L-ascorbic acid, dopamine hydrochloride, etc. for Pt/Nafion(1/2)PPD sensors[103]) reduces sensor selectivity and therefore accuracy. A detailed review of various electrochemical NO sensors is discussed by Privett et al.[102]. Additionally, a few of the porphyrin-based electrochemical sensors developed by Malinski and Taha[84] and Vergnani et al.[104] have been used to measure NO release from healthy and diseased endothelial cells or in tissues.

*Organic-based NO sensors*. Molecular probes potentially offer sub-cellular selective detection of an analyte, high selectivity, and the possibility to combine NO sensing with other molecular probes. Regarding NO sensing, many molecular probes, such as those developed by the Nagano group, are organic fluorophores conjugated to an ortho-phenylenediamino group[105]. In the absence of NO, the fluorescence of the fluorophore is quenched by photoinduced electron transfer (PET). PET is due to the high energy of the electrons in the free amine group, which allows this electron to relax in the lowest SOMO orbital of the probe's

excited state, followed by relaxation of the electron in the high-energy SOMO or the probe's excited state, back into the temporarily empty free amine orbital. In the presence of dioxygen, NO and the phenylenediamine probe react chemically to form an electron-poor triazole ring. The non-bonding electron pairs of this triazole ring are much lower in energy, as a consequence of which they cannot quench the excited state of the nearby fluorophore. In phenylenediamine-based molecular probes for NO the PET quenching process is blocked upon forming the triazole ring in the presence of NO, which enhances fluorescence emission, compared to the diamine, thereby allowing NO sensing[106–108]. The nature of the fluorophore allows for fine-tuning the excitation and emission properties of the sensor; the type and size of the linker separating the fluorophore from the diamine moiety influences the difference in brightness between the on and off states; and the overall molecular properties of the probe, such as its charge, hydrophobicity, or molecular shape, influence its cellular uptake and intracellular localization.

Figure 4b shows various NO sensors based on the o-phenylenediamino group, which differ by the nature of the fluorophore and its localization inside the cell. Amine-based NO sensors are prone to interference from protonation[105–107]. The pH-dependence of bodipy-amine NO sensors was discussed in detail by the Nagano group wherein at pH 7 and above, the fluorescence was quenched due to the accelerated PET process. At high pH, the triazole loses a hydrogen atom and forms the triazolate with a higher HOMO energy turning on the PET process which was turned off in triazole due to the lower-lying HOMO energy compared to that in the amine-functionalized fluorophore. According to the Rehn–Weller equation, the free energy change of the PET process is determined by both the electron-donating ability of reactive triazole sites and by the reduction potential and the excitation energy of the fluorophore. Therefore, modification of the fluorophore to change the reduction potential and the excitation energy can facilitate the synthesis of a pH-independent amino-based NO sensor[109]. Other organic functional groups employed as the NO recognition site include dihydropyridine (Hantzsch ester)[110,111] and aromatic secondary amines[112–114]. Though these diamine probes are by far the ones that have been mostly used in the literature, they do not directly detect NO, but an intermediate $N_2O_3$ is a nitrosating agent for diamine as shown in Eq. (3). The rate constant for Eqs. (1) and (2) and nitrosation of morpholine by $N_2O_3$ are $6.3 \times 10^6 \, M^{-2} \, s^{-1}$, $1.1 \times 10^9 \, M^{-1} \, s^{-1}$ and $6.4 \times 10^7 \, M^{-1} \, s^{-1}$, respectively, indicating that Eq. (1) is the rate-determining step[115–117].

$$2NO + O_2 \rightarrow 2NO_2 \quad (1)$$

$$NO + NO_2 \rightarrow N_2O_3 \quad (2)$$

$$N_2O_3 + R_2NH \rightarrow R_2NNO + NO_2^- + H^+ \quad (3)$$

*Metal complex-based NO sensors.* On the other hand, Lippard et al. and others have focused on the development of molecular probes for NO based on transition metals. In the absence of NO, the fluorescence of these probes is quenched by the coordination of nitrogen- or oxygen-based ligands to a paramagnetic transition metal ion, typically Cu(II). In the presence of NO, the fluorescence is recovered[118]. Lippard's group provided for example direct fluorescence detection for NO in living cells using Cu(II)–fluorescein (FL) complexes, where the FL molecule comprises an 8-aminoquinaldine ligand attached to the 40 position of the fluorescein xanthene ring. The formation of a 1:1 Cu$^{II}$:probe complex (CuFL) resulted in a dim fluorescent species. Reaction with NO generated a bright nitrosylated fluorophore FL-NO. The reaction is presumably occurring through a NO-mediated

reduction of Cu(II) to Cu(I) followed by nitrosylation of FL and finally dissociation of Cu(I) halide as shown in Fig. 4d, thereby recovering emission[118,119]. Sensors based on other metal ions, including Co(II), Fe(II), Ru(II) and Rh(II), have also been developed recently[120]. Other metal-based sensors use colour changes upon axial binding of NO to Fe(II) porphyrins and Fe(III) corroles (Fig. 4d); this might also be a promising alternative for optical NO detection[6,121]. A major advantage of these sensors compared to those based on the o-diaminophenylene group, is that they do not require dioxygen to be present to detect NO, as they solely react with NO. Being able to detect NO in the absence of oxygen would be beneficial for investigating ED under hypoxic conditions, as hypoxia can lead to uncoupling of the eNOS enzyme, leading to a decrease in NO production[122]. Therefore, developing oxygen-independent NO sensors are of great importance.

*Nanoparticle-based NO sensors.* Recent advances in the field of NO sensing have also shown the development of nanoparticle-based sensor for NO including those based on upconverting nanoparticles, quantum dots, and plasmonics-based nanoparticles. These functional bio-nanomaterials are advantageous over other materials for sensing NO at the plasma membrane[123]. Their smaller size makes them ideal for interfacing non-invasively with the plasma membrane, peptides[124–126], and cholesterol derivatives[127,128]. Nanoparticles also possess various optoelectronic properties (FRET, charge transfer, etc.) to serve NO sensing at the membrane. On the other hand, the tunability of circulation time and clearance properties in animal models makes them good candidates for NO sensing in vivo[129]. For example, the Li group developed an upconversion nanoparticle (UCNP)-based NO probe for the ratiometric measurement of NO in cells[130]. The probe consists of a UCNP core surrounded by a shell of meso-porous $SiO_2$, loaded with rhodamine B-derived molecules (the NO-reactive molecule). NO sensing is achieved via luminescence resonance energy transfer (LRET) between rhodamine B and the UCNP. Reaction with NO shows a recovery of the strong absorption of rhodamine B at 500–600 nm. Quantitative measurement of NO was achieved through the intensity ratio of rhodamine B emission to upconverted emission from the UCNP, $I_{656}/I_{540}$. The Da-Wei Li group developed gold nanoparticles modified with o-phenylenediamine (OPD)- based NO sensor to detect the level of the endogenous NO in murine macrophages (RAW 264.7)[131]. In presence of NO, OPD reacts with NO to form a triazole, resulting in SERS variations of the AuNPs/OPD nanoprobes. The intensity ratio, $I_{789}/I_{585}$ was used as indicator for NO level, as the peak at 789 cm$^{-1}$ was strongly dependent on the NO-triggered reaction whereas the peak at 585 cm$^{-1}$ was nearly unrelated to the presence of NO. The Li group also developed Au-Ag alloy/porous-$SiO_2$ core/shell nanoparticle-based NO probe for ratiometric analysis of NO in HeLa cells[132]. A 3,4-dia-mino-benzene-thiol (DABT) was self-assembled on the Au-Ag alloy nanoparticle's surface to get the NO probe and a strong scattering peak at 804 cm$^{-1}$ appeared after treatment with NO. The researchers also showed nitrogen oxides did not interfere with the sulfhydryl group in Au-S bonds in the probe indicating good stability and excellent selectivity of the SERS nanoprobe. The Huang group reported a red-emitting Fe(III)-bound dithio-carbamate functionalized CdSe-ZnS quantum dot (QD) NO sensor showed a subsequent increase in QD photoluminescence upon NO exposure due to decreased FRET from the QD donor[133]. The NO probe showed a limit of detection (LOD) for NO at ~3.0 μM. Liu et al. developed a modified hyperbranched polyether (mHP) nanospheres with fluorescent CdSe QDs encapsulation for NO detection and release[134]. The QDs-mHP-NO nanosphere releases NO molecules when placed in aqueous

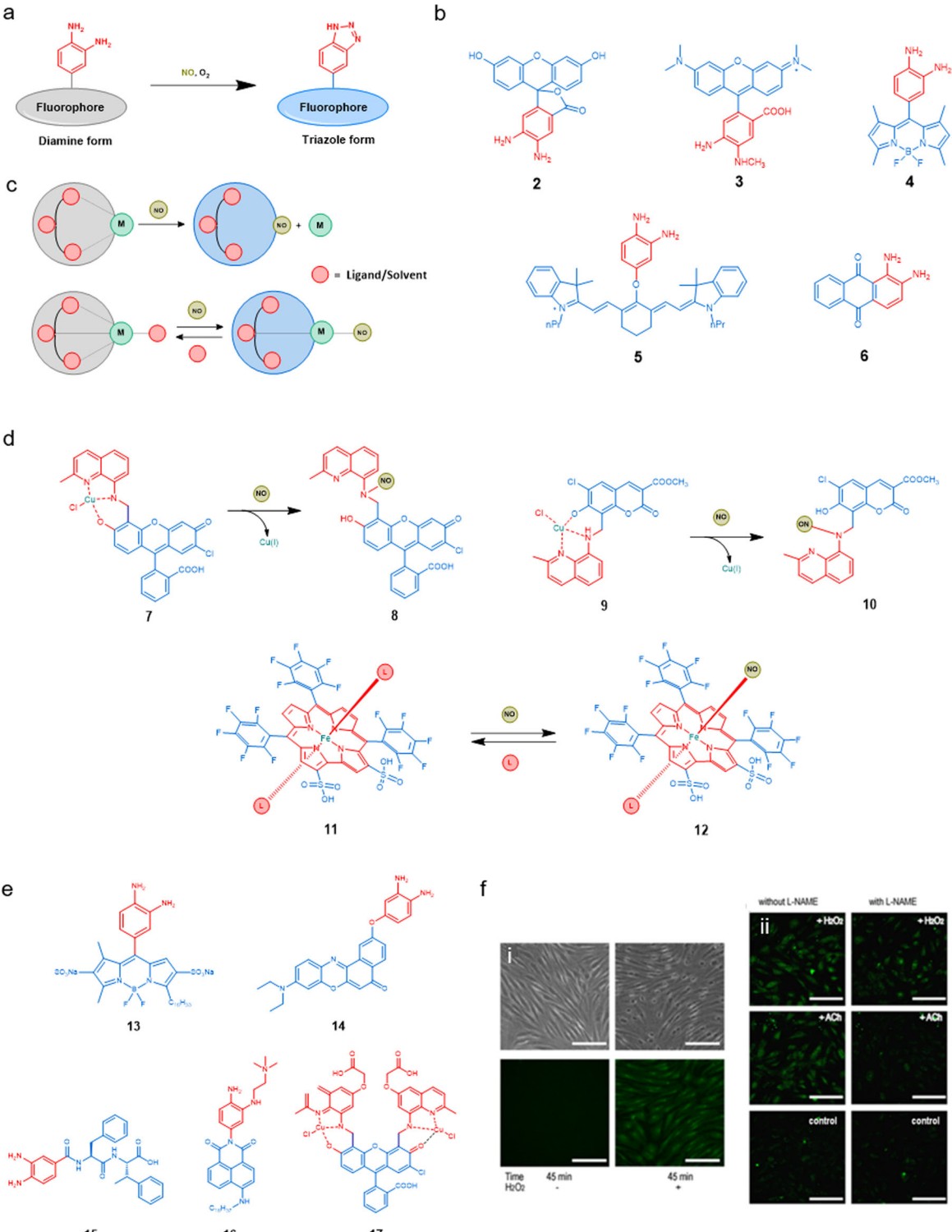

**Fig. 4 Chemical structures of chosen NO sensors and example for NO sensing from endothelial cells. a** General scheme for the transformation of phenylene diamine-functionalized fluorophore to a triazole-functionalized analogue. The grey colour represents a fluorophore that emits poorly because it is quenched by PET, while the blue colour represents a fluorophore that has recovered a strong emission. **b** Molecular formulae of a selection of organic nitric oxide sensors based on the phenylenediamine motif[119,189,190]. **c** General sensing method for NO by metal-based fluorescent probes. **d** Molecular formulae of selected examples of transition metal-based NO sensors[191,196]. **e** Selected nitric oxide sensors useful for sensing endothelial dysfunction. **f** Detection of NO produced by endothelial cells in vitro using Lippard's molecular probe **17**. (i) NO detection in porcine aortic endothelial cells (PAECs); Left: 45 min incubation of **17** (20 μM). Right: 45 min incubation of **17** (20 μM) and $H_2O_2$ (150 μM). Top: bright-field images of cells. Bottom: fluorescence images of cells. Scale bar is 50 μm. (ii) Detection of NO with **17** in Human Coronary Artery Endothelial Cells (HCAECs), with or without NO-inhibitor (L-NAME). Shown are the fluorescence images after 45 min co-incubation of the probe **17** (2 μM) with $H_2O_2$ (150 μM), L-NAME (100 μM), and/or Acetylcholine (ACh) (10 μM) according to scheme. Scale bar is 75 μm (Reprinted with permission from ref. [146]).

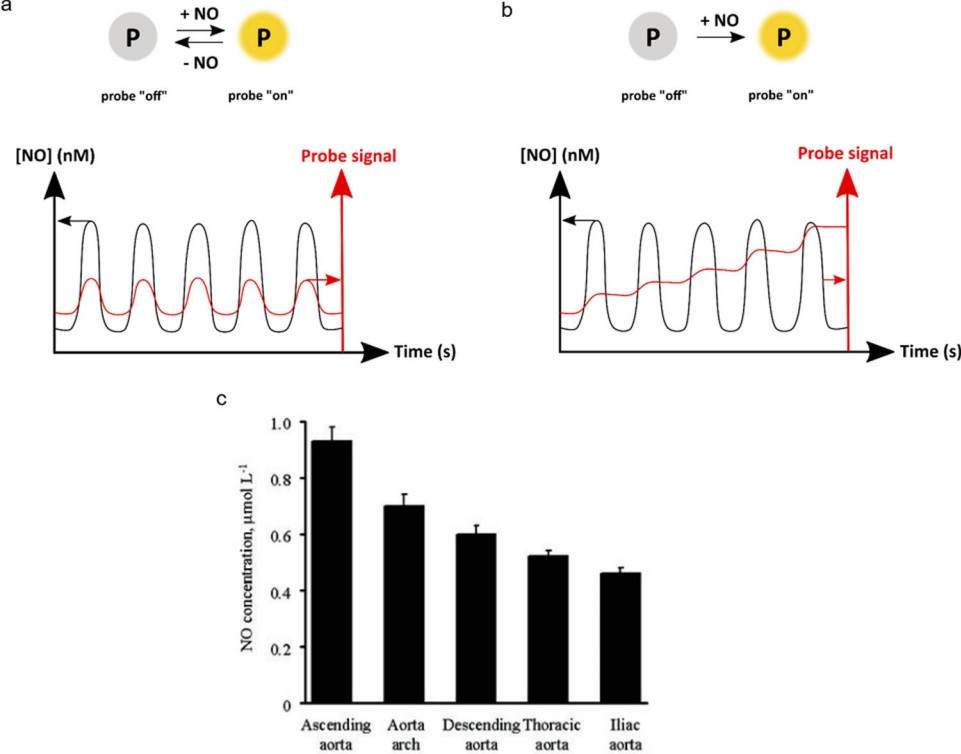

**Fig. 5 Schematic representation of reversible vs. irreversible NO sensor and example of NO level found in aortic endothelial cells. a** Reversible vs. **b** irreversible NO sensing probes, and relation of reversibility to time-dependent NO level detection. **c** Maximal NO concentration measured in vitro in endothelial cells of aorta (Reprinted with permission from ref. [51]). All NO levels measured with an electrochemical NO sensor.

buffer resulting in an increase in QD photoluminescence, whereas introducing NO to the QDs-mHP nanosphere increases the fluorescence quenching. The nanoprobe also showed high selectivity in the presence of other ROS and RNS like $H_2O_2$, $ClO^-$, $ONOO^-$, $NO_2^-$ and $NO_3^-$. Other nanoparticle-based systems for NO sensing developed recently include probes based on Ce-based metal−organic framework[135], aminoguanidine hydrochloride and citric acid-based carbon-dots[136], 4-(((3-amino-naphthalen-2-yl)amino)methyl)benzoic acid functionalized CdTe/CdS/ZnS quantum dots[137], electrochemically active gold nanoparticle-modified tungsten oxide ($WO_3$) nanoflakes[138], or Au nanoparticles that undergo azide–alkyne click chemistry-mediated aggregation-based NO sensor[139].

**NO sensors for ED**. Properties for a suitable sensor for NO have been discussed in section "Towards molecular probes for the sensing of ED" as well as some desired properties in section "Challenges in NO sensing". Taking those into consideration, over the past 5 years many biocompatible NO sensors have been developed, which are good candidates for sensing ED. In 2016, the Wang group[140,141] reported a BODIPY-based NO fluorescent probe (**13**, Fig. 4e) using o-phenylenediamine as the NO trapping group. The excellent amphiphilic property of the probe facilitated the probe to anchor onto the cell membrane, which helped in imaging extracellular NO released from the cell. Liang and coworkers[142] reported a water-soluble and biocompatible fluorescent probe **15**, in which the o-phenylenediamine was attached to two phenylalanines. This fluorescent probe exhibited both excellent water solubility and biocompatibility for intracellular study. In 2019, Zheng and coworkers[143] reported a novel fluorescent probe (**14**, Fig. 4e). This probe emitted at a comparatively long wavelength (658 nm), which offers longer tissue penetration

depths. In 2018, Wang and coworkers[144] prepared a two-photon fluorescent probe (**16**, Fig. 4e). The hydrophilic quaternary ammonium unit makes it highly water-soluble and the hydrophobic lipid tail was introduced to anchor the probe on the cell membrane. Lippard's group has also synthesized second-generation probes (**17**, Fig. 4e)[145]. When the membrane-permeable probe, **17**, enters the cells, the ester moiety is hydrolysed by cytosolic esterases to yield a carboxylate appended probe and the membrane-impermeable and negatively charged acid derivative is trapped within the cell[146].

**Irreversible vs. reversible NO sensors**. One major aspect that still needs to be addressed in NO sensing is the reversibility and reaction rate of NO with ortho-diamine sensors, considering the time-dependent character of NO production in the endothelium. A *reversible* sensor is a sensor that not only binds to NO but also releases it (Fig. 5a). Ideally, if the concentration of a probe in cells is small enough, sensing will not significantly modify the spatiotemporal NO concentration patterns, and it might be possible to follow in time the NO concentration without modifying it, as lowering NO concentrations would lead to a release of NO from the sensor. By contrast, *irreversible* NO probes react irreversibly with the analyte (Fig. 5b), which disappears from the biological medium. Here as well, low probe concentrations are required to keep the minimal influence of the probe on the biological effects of NO. However, with irreversible sensors, it is only possible to *integrate* in time the temporal variation of NO concentration. Indeed, such probes cannot release NO even if the local NO concentration becomes lower for some time. From an in vivo studies perspective, this scenario may be sub-optimal, as it is not beneficial to disturb the biological NO level when trying to quantify it. Thus, reversible NO sensors are in principle more

**Table 1 Nitric oxide sensors discussed in this review.**

| ss | Name | $\lambda_{ex}$, nm | $\lambda_{em}$, nm | Selectivity to NO against $NO_2^-$, $NO_3^-$, $ONOO^-$, $ClO^-$, $H_2O_2$, $OH^-$, $O_2^-$ | Limit of detection | Cells tested | Time taken to reach fluorescence maximum | Ref. |
|---|---|---|---|---|---|---|---|---|
| 1 | ER-Nap-NO | 440 | 538 | Selective | 3.3 nM | HeLa | ~6 min | 94 |
| 2 | DAF-2 | 486 | 513 | Selective; $ClO^-$, $OH^-$ not tested | 5 nM | RASMC, RAW 264.7 | ~30 min | 105 |
| 3 | DAR-4M | 543 | 572 | – | 7 nM | BAEC | – | 189 |
| 4 | DAMBO | 495 | 505 | Selective; $ONOO^-$, $ClO^-$, $O_2^-$ not tested | – | – | >30 min | 109 |
| 5 | DAC-P | 767 | 782 | – | – | Rat kidney | ~3 min | 108 |
| 6 | DAA | 540 | 643 | – | 5 mM | RAW 264.7 | – | 190 |
| 7 | Cu(FL$_5$) | 499 | 520 | Selective | 5 nM | RAW 264.7, SK-N-SH | >5 min | 118 |
| 9 | Cu(II)CB5 | 410 | 448 | Selective | – | P. aeruginosa, RAW 264.7 | – | 191 |
| 11 | Iron(III) corrole | 403 | NA | – | – | – | NA | 121 |
| 13 | DSDMHDA | 508 | 526 | Selective | 2 nM | RAW 264.7, ECV-304 | ~1 min | 140,141 |
| 14 | TTNO | 591 | 658 | Selective | 9 nM | HeLa, RAW 264.7 | ~35 min | 143 |
| 15 | OPD-FF | 275 | 367 | Selective | 6 nM | HepG2 | ~3.5 min | 142 |
| 16 | Mem-NO | 435 | 538 | Selective | 74 nM | HUVEC, PC12 | >30 min | 144 |
| 17 | Cu$_2$FL2E | 496 | 526 | Selective | 35 nM | RAW 264.7, SK-N-SH | ~20 min | 145,146 |

All the sensors are turn-ON sensors.
*NA* not applicable.

attractive than irreversible sensors, at least when the analyte concentration depends on time. Organic-based NO probes based on diaminophenyl groups have been widely employed for NO imaging in cells, tissues, and organs. Still, despite their undeniable utility, they react irreversibly with NO to form the triazole ring (see above); they hence *integrate* the NO time evolution. In contrast, inorganic copper-based fluorescent probes and iron-corroles bind reversibly with NO; in principle, they are hence better suited for following in-time NO concentrations.

**Time evolution of NO concentration**. Many excellent NO sensors have been described in the literature. Still, only the electrochemical-based sensors can give some information regarding the time evolution of NO concentration in living endothelial cells. Studies conducted by Balbatun et al. on endothelial cells from the iliac artery of rats revealed that upon treatment with calcium ionophore (A23187, $1\,\mu mol\,L^{-1}$), a healthy endothelial cell releases NO at the rate of $1200 \pm 50\,nmol\,L^{-1}\,s^{-1}$ with a peak concentration of $430 \pm 15\,nmol\,L^{-1}$ reached after $600 \pm 20\,ms$. Whereas in a diseased endothelial cell, NO is released at the rate of $460 \pm 10\,nmol\,L^{-1}\,s^{-1}$, and a NO peak at a concentration of $140 \pm 15\,nmol\,L^{-1}$ was reached after 900 ms[51]. Similarly, many other agonists like acetylcholine and cicletanine were also used to induce NO release and to study the time evolution. And it was observed that the rate of NO release and NO peak concentration varied with different agonists. The peak NO concentration achieved in the presence of calcium ionophore ($595 \pm 30\,nM$), acetylcholine ($390 \pm 20\,nM$) and cicletanine ($160 \pm 8\,nM$) are all different. Moreover, the kinetics of NO release after stimulation with acetylcholine are distinctively different from those observed after stimulation of NO release by calcium ionophore[147]. Further studies for the concentration of NO released by normal endothelial cells from different parts of the aorta of rats indicated that the dynamics of NO release differs significantly in different locations of the cells within the aorta (Fig. 5c)[51]. With all these facts in mind, it is very hard to come up with an accurate value for the rate or peak value of NO release from endothelial cells. Therefore, to improve the measurement of time dependence for NO concentration, the question of the kinetics of NO reaction

with the probe should also be taken into consideration. In principle, being able to sense the tie variations of the NO concentrations requires the reaction with NO to occur at a time scale that is shorter than the time characteristic for the biological NO production cycle. All sensors discussed above have different detection times for NO. An overview of the kinetic data available in the literature is provided in Table 1. However, no clear-cut values for the time scale of a biological cycle for NO production by endothelial cells have been described yet. A similar uncertainty exists for the real physiological concentration of NO. Two decades ago, a NO concentration of about $1\,\mu M$ measured using electrode-based sensors, seemed reasonable. Since then, numerous other NO sensors have been developed. Nowadays, evidence from these new methods points to physiological NO concentrations between 100 pM and 5 nM[45]. Overall, the lack of agreement between different sensors published by different groups, and the lack of time-dependent data, make that the field of NO sensing in living cells, particularly in endothelial models, is still widely open.

## Calcium chemosensors

**Calcium and ED**. In the literature, ED is usually defined as a decrease in NO released by endothelial cells, particularly in response to an external stimulus like shear stress. Such a distinction, however, does not allow specifically to detect ED with a real sample. Following such a criterion is indeed difficult, as it would be necessary to compare diseased cells with healthy ones, and to provide a parallel investigation of the variation in time of the NO levels in the two types of cells, which is usually not done. In addition, this approach does not allow to move from in vitro to in vivo. Instead of measuring a second healthy cell line, one could also measure a second parameter inside the endothelial cells of the study, at the same time as the NO level is measured. When considering the biological pathway for NO production, studying the $Ca^{2+}$ levels concomitantly to NO levels could be highly beneficial for understanding their interactions in endothelial cells. Calcium sensors have been under study for a long time due to their importance in cellular signalling, and therefore, several reviews are available on that topic[148–151]. In this part, we discuss more precisely calcium sensing in relation to ED.

### Selectivity and sensitivity

*Binding affinities.* $Ca^{2+}$ sensing is usually based on reversible binding to a calcium-binding chelate. The values of the intracellular $Ca^{2+}$ concentration have been previously determined by different methods; Fig. 6 describes several stimuli used in the literature to influence $Ca^{2+}$ concentration in cells[46,152–156]. A review from 2003[157] summarizes that after stimulation cytosolic level of free $Ca^{2+}$ increases to 0.5–1 µM in two steps. First, $Ca^{2+}$ is released from the endoplasmic reticulum and then, from outside the cell, by concentration gradient it flow into the cell via the transient receptor potential (TRP) channels. Thus, for healthy EC, cytosolic $Ca^{2+}$ level reaches ~1 µM in the presence of shear stress. Unfortunately, we were not able to find published values of $Ca^{2+}$ concentration in dysfunctional EC, so only hypotheses can be made. According to Du et al.[27] the $Ca^{2+}$ influx in the cytosol of ECs, even in the case of ED, is not blocked. Hence, we can suggest that $Ca^{2+}$ concentration in the cytosol of EC, even in the case of ED, still increases in the presence of shear stress. Thus, for further

ED sensor design, it might be reasonable to consider that $Ca^{2+}$ levels also increase up to 1 µM in the cytosol of dysfunctional EC upon external stimulus.

Usually, $Ca^{2+}$ chemosensors can be divided into two parts, a $Ca^{2+}$-chelating moiety and a luminophore—often a fluorophore. The calcium-binding ligand is mostly characterized by its selectivity towards other biologically relevant metal ions (for example, $Zn^{2+}$ and $Mg^{2+}$), and its sensitivity to $Ca^{2+}$, in which for a range of $Ca^{2+}$ concentrations, the luminescence will vary. The sensitivity might be expressed as a ratio $\Delta F/F_0$, where $\Delta F$ is a difference in the emission of the complex CaL ($F_{max\,or\,min}$) vs. free sensor L ($F_0$):

$$\Delta F/F_0 = \frac{F_{max\,or\,min} - F_0}{F_0} \qquad (4)$$

Contrary to diamine-based NO sensors, which show slow NO binding and are hence usually under kinetic control (Table 1), the binding of $Ca^{2+}$ ions to chelates is usually considered as being fast. Hence, calcium molecular probes are usually considered as being under thermodynamic equilibrium. As a consequence, the main parameters describing analyte binding to the sensor is its association constant $K_a$ (in $M^{-1}$) or dissociation constant $K_d = 1/K_a$ (in M). The sensitivity of $Ca^{2+}$-binding chemosensors is strongly dependent on $K_d$. $K_d$ is defined by Eq. 5, in Box 2, and its value numerically represents the $Ca^{2+}$ concentration for which half of the chelate is bound to $Ca^{2+}$. Additionally, an increase in $Ca^{2+}$ concentration typically induces NO production. Thus, a perfect calcium chemosensor should not disrupt the biological process of NO production, and its intracellular concentration should remain low, compared to biological $Ca^{2+}$ concentrations. From simple $K_d$ calculations (Box 2), it should be highlighted that the higher the $K_a$ is, the lower amount of probe is needed to detect a given intracellular ions concentration, and the lower amount of $Ca^{2+}$ cations are involved in the measurements, which minimizes the influence of calcium sensing on the biological system.

Another important characteristic of a $Ca^{2+}$-chelating moiety is its selectivity to $Ca^{2+}$. Optimal selectivity is obtained when the binding is not too tight. In particular, when working within a cell the presence of other metal cations in the cytosol should be taken into account. The metal cations with the highest concentration in the cytosol are $Na^+$, $K^+$, $Ca^{2+}$, $Mg^{2+}$, and to a lesser extent, $Zn^{2+}$[2,197]. Of these cations, the most similar to $Ca^{2+}$ are $Mg^{2+}$

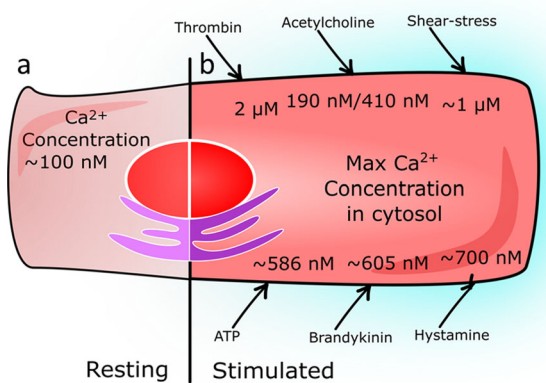

**Fig. 6 $Ca^{2+}$ intracytosolic concentration maximum in different studies.**
**a** Resting cell characterized by low $Ca^{2+}$ levels in the cell cytosol (100 nM). **b** Stimulated cell has increased $Ca^{2+}$ levels in the cell cytosol depends on stimulus. Stimulus ($Ca^{2+}$ concentration): thrombin (2 µM), acethylcholine (410 nM), shear–stress (1 µM), adenosine triphosphate (ATP) (586 µM), Brandykinin (605 nM), Hystamine (700 nM)[46,152–156].

---

**Box 2 | What is a dissociation constant of an ideal calcium sensor for endothelial dysfunction?**

If in a cell the initial calcium ion concentration is $[Ca^{2+}]_o = 1$ µM, and if after the addition of a calcium ion sensor L at a concentration $L_0$ we would like to keep the $Ca^{2+}$ concentration after binding to the sensor mostly undisturbed, e.g., $[Ca^{2+}] = 0.99$ µM, then the concentration of the sensor-calcium complex CaL should be 0.01 µM (1%). Let us consider here two sensors $L_1$ and $L_2$ with different $K_d$, $K_{d1} = 0.1$ µM and $K_{d2} = 10$ µM. To calculate the sensor concentration, we need to add to the cell to detect 1 µM of calcium ions, one should consider the following simple calculation:

$$Ca^{2+} + L \rightleftarrows CaL$$
$$K_d = \frac{[Ca^{2+}][L]}{[CaL]} \qquad (5)$$

Sensor $L_1$
$$0.1\,µM = \frac{0.99 \times [L]}{0.01}$$
$$[L] = 1.01\,nM$$
$$L_{0,1} = [L] + [CaL] = 1.01\,nM + 10\,nM = 11.01\,nM$$
$$\frac{[CaL]}{L_{0,1}} = \frac{10}{11.01} \times 100\% = 91\%$$

Sensor $L_2$
$$10\,µM = \frac{0.99 \times [L]}{0.01}$$
$$[L] = 101\,nM$$
$$L_{0,2} = [L] + [CaL] = 101\,nM + 10\,nM = 110\,nM$$
$$\frac{[CaL]}{L_{0,2}} = \frac{10}{110} \times 100\% = 9\%$$

In other words, if the sensor $L_2$ binds 100 times more loosely to calcium than $L_1$, one needs to add 10 times more sensor ($L_{0,2} = 110$ nM) in the cell to have the same (low) amount of $Ca^{2+}$ bound. Of course, higher sensor concentrations are not ideal because of the potential toxicity to the cells.
Also, with the tighter-bound sensor $L_1$, the fraction of the bound sensor is 10/11.01 = 91%, while with $L_2$ it is 9%. Thus, for the first sensor, $\Delta F/F_0$ (Eq. 4) is higher than for the second one, which means that the imaging contrast with $L_1$ is higher. Hence, in theory, it is better to have a tight binding ($K_d$ lower than the $Ca^{2+}$ concentration to measure) to minimize the amount of sensor needed and its toxicity, while maximizing contrast in imaging.

**Table 2 Calcium sensors discussed in this review.**

| Number | Name | $K_d$, μM | Selectivity to Ca$^{2+}$ against Mg$^{2+}$, Zn$^{2+}$ and Na$^+$ | $\lambda_{em}$, nm | pH | Stability in cell environment | Turn-on sensor (ON)/Turn-off (OFF) | Ref. |
|---|---|---|---|---|---|---|---|---|
| 18 | BAPTA sensor family | 0.15; ~0.70[a] | Selective | Fluorophore dependent | Cytosolic[d] | Stable | ON | 192 |
| 19 | – | 256[b] | Selective | 488 | – | – | ON | 161 |
| 20 | BCaM | 90 | Selective | 450 | Cytosolic[d] | Stable | ON | 193 |
| 21 | Bichromophore | 14.8 | Selective | 436 | – | – | ON | 194 |
| 22 | Calix[4]arene | 121[b] | Selective | 510 | – | – | ON | 162 |
| 23 | – | 2400 | Selective | 550 | 8.1 | – | ON | 163 |
| 24 | PIAQ–AC 5a | 0.309[b] | Not selective | 515 | – | – | ON | 164 |
| 25 | PIAQ–AC 5b | 0.069[b] | Not selective | 515 | – | – | ON | 164 |
| 26 | – | 0.0192 | | 485[c] | – | – | | 165 |
| 27 | DMK | 0.0144 | Selective | 390 | 4 | – | ON | 195 |
| 28 | ISB | LOD = 0.3 | Selective | 483 | 7.2 | – | ON | 166 |
| 29 | MPFCP-1 | 0.44 | Selective | 525 | Cytosolic[d] | Stable | ON | 167 |
| 30 | MPFCP-2 | 1.21 | Selective | 525 | Cytosolic[d] | Stable | ON | 167 |
| 31 | BODIPY-BAPTA | 0.501[b] | Selective | 508 | 7.2 | Stable | ON | 168 |
| 32 | JF$_{549}$-BAPTA | 0.31 | Selective | 569 | Cytosolic[d] | Stable | ON | 169 |
| 33 | CaSIR-1 | 0.58 | Selective | >650 | Cytosolic[d] | Stable | ON | 170 |
| 34 | – | 5120 | Selective | 655 | – | – | OFF | 171 |
| 35 | Di-Aza-crown-cNDI | 0.11[b] | Selective against Mg$^{2+}$ and Na$^+$ | 655 | – | – | ON | 172 |
| 36 | CaRB | 1.22 | Selective | 631 | 7.2 | 7.2 | ON | 173 |
| 37 | Calix[4]arene-BDP | – | Selective against Mg$^{2+}$ | 507 | – | – | OFF | 180 |

*FL* fluorescence, – not mentioned in the original article.
[a]$K_d$ with the presence of competitive ions such as Mg$^{2+}$.
[b]Recalculated from reported data.
[c]$\lambda_{em}$ before adding Ca$^{2+}$ ions.
[d]Sensor was tested in vitro, pH is in the range 7.2–7.4.

and Zn$^{2+}$. The main difference between these ions is their ionic radius (0.95 Å for Ca$^{2+}$, 0.6 Å for Mg$^{2+}$, and 0.65 Å for Zn$^{2+}$)[197]. Maximizing the difference in dissociation constant of the sensor, $K_d$, between the larger Ca$^{2+}$ ion on the one hand, and the smaller Mg$^{2+}$ and Zn$^{2+}$ ions on the other hand, is critical for calcium sensing in biology.

Currently, several chemosensors are commercially available for calcium imaging in cells. Most sensors have a 1,2-bis(*o*-aminophenoxy)ethane-*N*,*N*,*N*′,*N*′-tetraacetic acid (**18**, Table 2, Fig. 7a, BAPTA) moiety as the Ca$^{2+}$-binding part[192]. This chelator is a good example for illustrating the complexity of biological sensing. First, the $K_d$ of BAPTA binding to Ca$^{2+}$ changes in the presence of competitive cations. $K_d$ values of 150 nM and 700 nM towards calcium have been measured in the absence and the presence of 1 mM of Mg$^{2+}$ in cell culture media, respectively. The selectivity of this sensor is also limited. Since BAPTA is a chelate based on carboxylates, high concentrations of any cations will lead to binding. However, the calcium selectivity of BAPTA seems to be good at intracytosolic relevant cations concentrations: its light output is minimally influenced by 1 mM of Mg$^{2+}$ ions[198], 1 nM of Zn$^{2+}$ ions[199], 12 mM of Na$^+$ ions, and 140 mM of K$^+$ ions[197]. However, BAPTA shows fast decomposition even in freezer storage conditions[200]. Thus, the search for more sensitive, more selective and more stable chemical structures is still underway.

Other chemical structures have been described that are selective enough for calcium biosensing purposes. A selection is shown in Fig. 7a: the binaphthyl derivative **19**[161] reported by Wang et al., **20** (BCaM)[193], bichromophore **21**[194], anthraquinone-modified calix[4]arene **22**[162], the chemosensor **23**[163]. Unfortunately, proper selectivity does not guarantee good sensitivity, and the calcium-binding $K_d$ of these sensors are not low enough for in vitro investigations.

The development of new calcium-binding structures with higher sensitivity and higher calcium-binding affinities continues. For example, **24** and **25** sensors have $K_d$ values for Ca$^{2+}$ of 309 and 69 nM, respectively[164]. However, their aza-crown ether structure makes them suitable for different cations, and the

chemosensors are not very selective. The sensor **26**, reported by Kim et al.[165] has a $K_d$ value of 19 nM and was shown to be selective towards Ca$^{2+}$. The sensor **27**, presented by Kacmaz et al.[195] and **28**, reported by Saleh et al.[166] are sensitive enough to be used in biology. Unfortunately, the selectivity of both sensors was tested in the presence of only 10 μM of Na$^+$, Mg$^{2+}$ and Zn$^{2+}$ for **27** and 20 μM for the chemosensor **28**, which are lower than their levels in the cytosol. From these data, we cannot conclude whether or not these compounds are selective enough to be used for Ca$^{2+}$ imaging in biology. Overall, since BAPTA is one of the best Ca$^{2+}$ chelators available in the literature in terms of binding strength and selectivity, new Ca$^{2+}$ chemosensors with BAPTA as the chelating ligand appear regularly[167–170] (**29–33**, Table 2 and Fig. 7b), using different fluorophores to improve the photochemical characteristics of the molecular probe. Some of them are discussed below.

*Photophysical considerations*
Excitation and emission wavelengths: Since the fluorophore part of the molecule determines the wavelength of excitation and detection, certain fluorophores may have some benefits with respect to others. As mentioned above, different wavelengths provide different light penetration depths through tissues and blood[158]. The near-infrared (NIR, 700–950 nm) and infrared (IR, 1000–1350 nm) regions of the spectrum are sometimes called the "first" and "second" therapeutic windows, respectively[158,159]. They are preferred for designing calcium probes, although they are also more difficult to design (and sometimes to synthesize) than probes excited at lower wavelengths. With the light of higher energies (near ultra-violet to blue, i.e. 300–490 nm), the risk of causing damage to the cells during imaging, for example via a photodynamic effect (see above), is higher. Such phototoxicity often limits observation under a microscope for a long time[160]. Many of the chemosensors mentioned above have an emission wavelength, $\lambda_{em}$, in the green area (~485–550 nm)[161–168] (Table 2). Blood strongly absorbs light in this area[158], so such compounds cannot be easily used in vivo. Overall, to perform Ca$^{2+}$ sensing in vivo, red or NIR excitation and emission

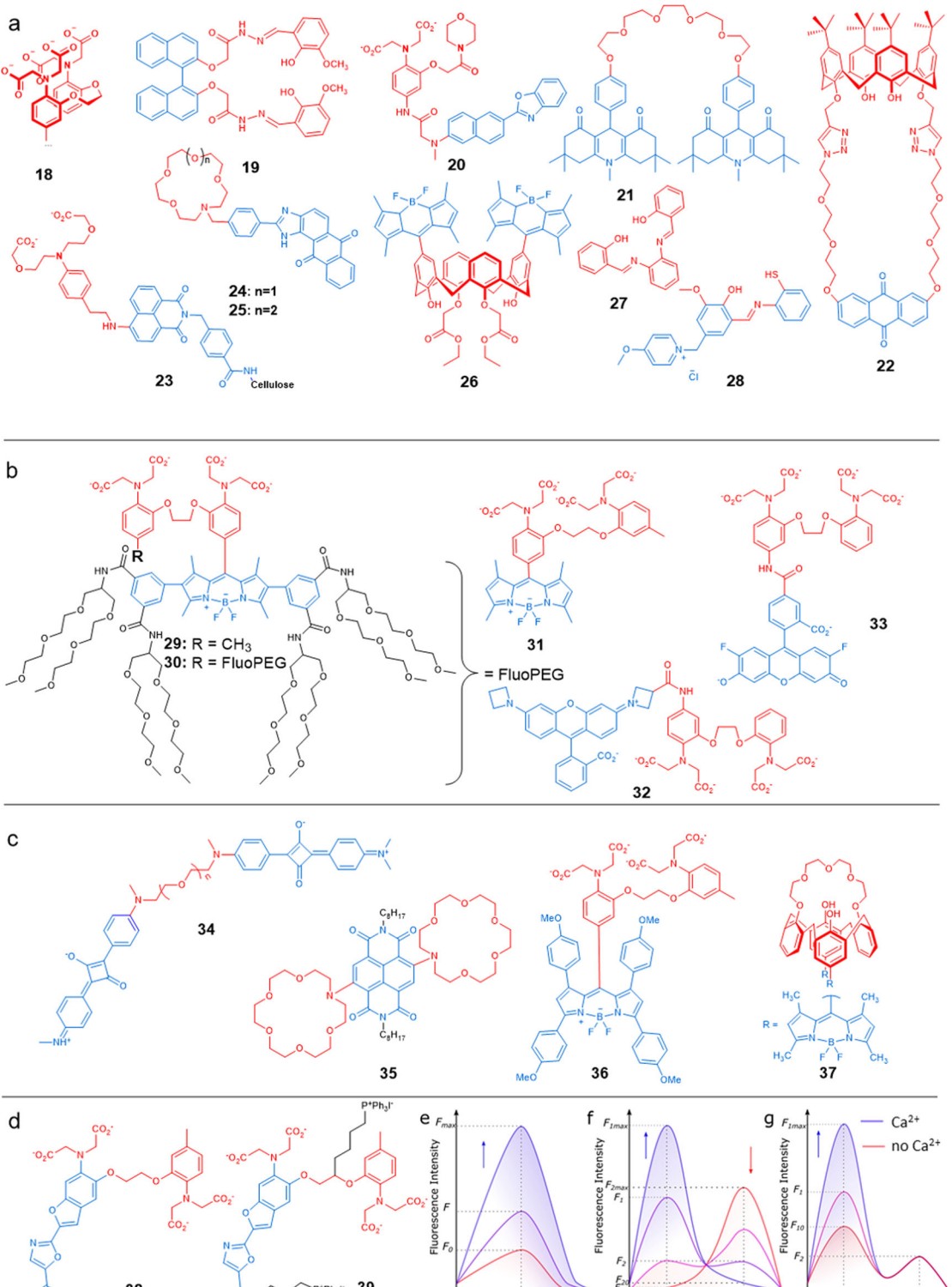

**Fig. 7 Chemical structures of chosen calcium sensors and graphical representation of intensometric, ratiometric and pseudo-ratiometric sensors.**
**a** Chemical formulae of a selection of published molecular probes for calcium ions:[161–166,192–195]. **b** Calcium chemosensors based on the BAPTA chelating unit[167–170]; **c** Calcium ions sensors that are excited by, or emit, red light[171–173,180]; **d** Chemical formulae of a selection of ratiometric chemosensors **38**, **39**[176]. **e–g** Typical emission spectra of **e** an intensometric $Ca^{2+}$ chemosensors, **f** a ratiometric $Ca^{2+}$ chemosensors, and **g** a pseudo-ratiometric chemosensors, upon gradual addition of calcium ions.

wavelength (>560 nm) is preferred, which incites research towards new calcium probes. For example, sensor **32** (Fig. 7b, JF$_{549}$-BAPTA) provides excitation ($\lambda_{ex}$) and emission ($\lambda_{em}$) wavelengths of 546 nm and 569 nm, respectively[169] while **33**[170]

has an even higher $\lambda_{ex} = 620$ nm and $\lambda_{em} > 650$ nm. Chemosensors such as **34**[171] absorb and emit red light (Table 2 and Fig. 7c), but sometimes their $K_d$ is too high for use with biological purposes. Still, novel $Ca^{2+}$ chemosensors with high sensitivity

and selectivity towards $Ca^{2+}$, and that can be excited in the red region of the spectrum, have been recently reported, for example, di-Aza-crown cNDI **35**[172] and CaRB **36**[173].

Intensometric, ratiometric and pseudo-ratiometric photochemosensors: According to its photophysical behaviour, a $Ca^{2+}$ chemosensor might be included in one of the three following groups: (i) intensometric, (ii) ratiometric or (iii) pseudo-ratiometric chemosensors[174]. The type of sensing influences the relation between the light intensity and the $Ca_0^{2+}$ concentration to be measured (i.e. before the calcium ions bind to the sensor in the cell, Fig. 7d–g). These three types of responses are shortly described below.

First, the fluorescence intensity of intensometric sensors is proportional to the $Ca_0^{2+}$ concentration (Figs. 7e and S1a). The advantage of such chemosensors is their ability to extract $Ca_0^{2+}$ concentrations relatively easily from the emission data if the intracellular concentration of the sensor itself ($L_0$) and the fluorescence intensity are known. $Ca_0^{2+}$ is given by Eq. 6 (derivations of equations are given in the supplementary information).

$$Ca_0^{2+} = K_d \frac{F - F_0}{F_{max} - F} + L_0 \frac{F - F_0}{F_{max} - F_0}, \qquad (6)$$

where $F_{max}$ is the maximal fluorescence intensity of [CaL] complex, $F_0$ the fluorescence intensity of the $Ca^{2+}$-free sensor, and F the fluorescence intensity signal at a given $Ca_0^{2+}$ concentration.

However, often it is hard to know $L_0$ and $F$. In principle, the sensor intracellular concentration in vitro needs to be calibrated, but such calibration is usually impossible, as cellular uptake and localization cannot be quantified easily, while the emission of the probe depends on both concentrations in the bound complex ([CaL]) and the free ligand ([L]).

The problem of sensor concentration calibration in cellulo is usually solved by using a technique called ratiometric sensing. In such a technique, ratiometric or pseudo-ratiometric probes can be used. The absorption or emission spectra of such probes contain two bands, the ratio of which changes upon $Ca^{2+}$ addition. In an in vitro imaging experiment, one can usually directly determine the ratio between the two bands, which allows one to derive the ratio between the bound complex ([CaL]) and the free ligand ([L]) concentrations[175]. With so-called ratiometric probes (Figs. 7f and S1b), both emission bands change with $Ca_0^{2+}$. In such a case, the ratio between the two emissions of the two bands does not yet allow one to calculate $Ca_0^{2+}$ the final expression of this ratio still depends on $L_0$ (Eq. 7). For such sensors, determining the absolute local $Ca^{2+}$ concentration requires knowing the local sensor concentration. BAPTA-based chemosensors Fura-2 **38** and mt-fura-2 **39**[176] are typical examples of such $Ca^{2+}$ ratiometric probes (Fig. 7d). They usually allow for following relative $Ca^{2+}$ concentrations, for example, to follow the dynamics of $Ca^{2+}$ concentration flows in a cell.

$$Ca_0^{2+} = K_d \frac{R - R_{min}}{R_{max} - R} \frac{F_{2max}}{F_{20}} + L_0 \frac{F_{2max}(R - R_{min})}{F_{20}(R_{max} - R) + F_{2max}(R - R_{min})}, \qquad (7)$$

where R—ratio between fluorescence intensity of [CaL] complex and of $Ca^{2+}$-free sensor; $R_{max}$—ratio while all the sensor molecules been complexed with $Ca^{2+}$ ions; $R_{min}$—ratio between fluorescence intensities with absence of $Ca^{2+}$ ions in media.

For so-called pseudo-ratiometric sensors (Figs. 7g and S1c), one of the two emission bands is independent of calcium ion concentration. Pseudo-ratiometric sensors can be for example made by coupling two molecular systems into a single chemical

system: the first emitter should be an intensometric $Ca^{2+}$ sensor and the second a luminescent compound, the emission of which is shifted compared to the former and independent from the $Ca^{2+}$ ions concentration. In such probes, the relation between the sensor concentration $L_0$ and the intensity of the calcium-independent emission band can be calibrated. By measuring locally (i.e., for each pixel of the image) the intensity of the $Ca^{2+}$-independent emission, the local probe concentration can be obtained, and by comparing this emission intensity to that of the $Ca^{2+}$-dependent band the true local $Ca^{2+}$ ions concentration can be calculated. In other words, in pseudo-ratiometric sensors, $Ca_0^{2+}$ is independent of $L_0$ (Eq. 8), which allows quantifying local calcium ion concentrations in a cell.

$$Ca_0^{2+} = K_d \frac{F_1 - \frac{\alpha_1}{\alpha_2} F_2}{\frac{\beta_1}{\alpha_2} F_2 - F_1} + \frac{F_2}{\alpha_2} \times \frac{F_1 - \frac{\alpha_1}{\alpha_2} F_2}{\frac{\beta_1}{\alpha_2} F_2 - \frac{\alpha_1}{\alpha_2} F_2}, \qquad (8)$$

where $F_1$—$Ca^{2+}$-independent fluorescence intensity of the chemosensor; $\frac{\alpha_1}{\alpha_2}$—a proportion coefficient of fluorescence intensity of $Ca^{2+}$-free sensor to $Ca^{2+}$-independent fluorescence; $\frac{\beta_1}{\alpha_2}$—a proportion coefficient of maximal fluorescence intensity of [CaL] complex to $Ca^{2+}$-independent fluorescence.

To conclude on the different types of probes available, lifetime-based probes have been proposed to monitor analyte concentrations independently from the amount of the internalized probe using time-resolved spectroscopy, a technique called fluorescence- or phosphorescence lifetime imaging microscopy (FLIM or PLIM). This kind of sensing has been used a lot for dioxygen-sensing for example, which is often based on dynamic quenching of the triplet excited states of the probe by $O_2$ molecules[177]. In calcium sensing, however, changes of the emission intensity of the probe are usually based on "static quenching", i.e., interaction of the calcium analyte to be sensed with the sensor in its ground state. The molecular probe emission lifetime remains typically unchanged in this case, so that lifetime measurements as a measure of calcium concentrations are, in fact, rare[178].

*Quenching mechanism and contrast.* In the previous paragraphs, we discussed the question of the binding strength and selectivity of the calcium-binding chelating part and the photophysical properties of the fluorophore attached to the chelate. Usually, these different characteristics combine into calcium sensing via PET sensing (see Box 1)[179]. Similarly to NO sensing, in calcium sensors such as BAPTA-based derivatives shown in Fig. 7b, the calcium-free sensor molecule has an electron-rich group capable of quenching the dye emission via PET. Upon binding the sensor molecule to calcium ions, these electron-rich groups become engaged in coordination with the metal ion, which lowers the energy of the corresponding electron pair, quenches PET quenching, and hence recovers the emission of the sensor.

Such a sensing mechanism corresponds to a "turned-on" sensor, where the emission is recovered in the presence of the analyte. From Eq. (4), it is clear that for a "turned-on" sensor, a higher $\Delta F/F_0$ ratio leads to a larger difference between the emission of the $Ca^{2+}$-free ligand L and that of the CaL complex. The detection of a particular $Ca^{2+}$ concentration, both in vitro and in vivo, becomes more precise due to the increased signal-to-noise ratio. PET can also be used to switch off the light emission in the presence of the analyte. Those chemosensors are called "turned-off" chemosensors ($\Delta F/F_0 < 0$, i.e., the fluorescence of the CaL complex is lower than the fluorescence of the free probe L. Such "turned-off" probes are useful in areas where people need to detect the absence of an analyte. In biology measuring a fully "dark" state is typically challenging due to the countless photophysical processes happening at the same time in a cell.

For example, sensor 26[165], was able to detect nanomolar $Ca^{2+}$ concentrations, but its light output decreased in the presence of the analyte. In the absence of $Ca^{2+}$ the chemosensors 34[171] and 37[180] (Table 2 and Fig. 7c) show emission in the red part of the spectrum, but after calcium ions bind the emission intensity decreased, making these indicators "turned-off" probes. For in vitro and in vivo imaging, "turned-on" ($\Delta F/F_0 > 0$) sensors are usually preferable in particular because they offer the possibility to follow the dynamics of $Ca^{2+}$ concentration changes, and because of their higher signal-to-noise ratio.

*Kinetics.* Considering the time-dependent evolution of $Ca^{2+}$ fluxes in healthy and diseased EC, one could argue that not only the thermodynamics of the calcium-binding process but also the timescale for $Ca^{2+}$ binding and unbinding, hence the kinetics of the binding equilibrium, should be considered in the design of molecular probe for calcium sensing in ED diagnostics. If the $Ca^{2+}$ concentration in the cell cycles vs. time, it might be important to measure the variations of the $Ca^{2+}$ concentration with a probe that offers faster binding compared to the biological time variations of the $Ca^{2+}$ concentration. Indeed, such biological variations of [$Ca^{2+}$] might change drastically during the cycle of a healthy or diseased EC. Unfortunately, the time during which $Ca^{2+}$ concentration increases, how long its maximal concentration plateau persists, and how long it takes to decrease again to the resting state, are poorly known in EC. These kinetics are usually not considered in existing diagnostic tools. As for NO sensing, studies aimed at following in time calcium concentrations inside endothelial cells, are rare. A few kinetic studies with BAPTA-based sensors have been published in the 1980s and 1990s[181,182]. For example, it was shown that 38 (Fura-2) has a second-order rate constant for $Ca^{2+}$ binding of $k_{on} = 6.2 \times 10^8\,M^{-1}\,s^{-1}$ as defined by Eq. (12)[181].

$$\overset{k_{on}}{\underset{k_{off}}{Ca^{2+} + L \rightleftarrows CaL}}$$

$$r_{on} = k_{on}[Ca^{2+}] \times [L] - k_{off}[CaL] \tag{9}$$

$$r_{off} = -k_{on}[Ca^{2+}] \times [L] + k_{off}[CaL] \tag{10}$$

$$r = r_{on} - r_{off} \tag{11}$$

$$\text{then, } k_{on} = \frac{r/2 + k_{off}[CaL]}{[Ca^{2+}][L]} \tag{12}$$

$$\text{In case of equilibrium}: r = 0 \tag{13}$$

$$\text{Thus}: k_{on} = \frac{k_{off}}{K_d} \tag{14}$$

where $r_{on}$ and $r_{off}$ are the rates (in $M\,s^{-1}$) of the association or dissociation reactions; $k_{on}$ is an association rate constant (in $M^{-1}\,s^{-1}$) and $k_{off}$ is a dissociation rate constant (in $s^{-1}$).

Later this sensor was used for the measurement of $Ca^{2+}$ concentration increase in EC cytosol after stimuli using molecular drugs[183]. It was found that the concentration increases dramatically in 7–10 s after the drug addition to the cell media. Minute time scales were used in other studies[153,184,185] to show the dependence of $Ca^{2+}$ concentration vs. time. Today we know that $Ca^{2+}$ cytosolic concentration changes within seconds or even milliseconds[52]. Unfortunately, kinetic studies have not become a standard characterization routine for modern calcium sensors, so very little information is available in the literature about the most recent calcium probes.

**Conclusion on calcium probes potential for ED diagnostics.** A summary of the characteristics of the calcium probes discussed above is proposed in Table 2. Overall, despite the great structural diversity of these probes, only several of them, such as 29 (MPFCP-1), 30 (MPFCP-1), 31 (BODIPY-BAPTA), 32 (JF$_{549}$-BAPTA), 33 (CaSIR-1), 36 (CaRB) and 38 (Fura-2), have been tested in cells and could be used for ED cytosolic calcium concentration measurements in vitro. They are characterized by a dissociation constant close to 1 µM, a red or NIR absorption wavelength, a positive sensor response, and a good stability to the cell environment. While only 38 (Fura-2) and its derivatives have been used to investigate EC in models of ED or in diseased veins and arteries, we conclude from this analysis that sensors 33 and 36 have the highest potential in being utilized for ex vivo and in vivo research on ED. Moreover, though $Ca^{2+}$ in endothelial cells triggers the intricate pathway leading from shear stress to NO release, the effect of these calcium probes on NO release has not been investigated yet and remains unknown. In addition, very few articles report on the toxicity of these probes towards endothelial cells, and their toxicity in vivo remains unknown. Finally, considering that the cytosolic calcium concentration changes over time, kinetic studies should be included in the routine characterization of new $Ca^{2+}$ chemosensors to establish the time scales of the probe response.

**Outlook: detecting ED by combining NO and $Ca^{2+}$ sensing using logic gates**

As discussed above, $Ca^{2+}$ and NO concentrations are interdependent and time-dependent. This indication may be one of the reasons why diagnostic of ED is still in its infancy. To address the time dependence of [$Ca^{2+}$] and [NO], more precise kinetic studies will be necessary. On the other hand, understanding the interdependence between NO and $Ca^{2+}$ release in EC may make use of the concept of logic gates introduced decades ago by de Silva et al.[51,186–188]. In this approach, the evolution of the emission intensity of a molecular probe with experimental conditions can be re-interpreted in the form of a logic table by defining an emission intensity threshold $E_t$: when the emission intensity of the solution is higher than this threshold, the output of the logic gate is 1, and when it is lower than this threshold, for example due to some form of quenching, then the output of the logic gate is 0. The input of such a logic gate is also interpreted in terms of concentrations of the different analytes to sense. For example, when the concentration of a first analyte is higher than a threshold $C_{1,t}$, then the first input is considered 1, and when it is lower, it is 0. For the second analyte a second threshold $C_{2,t}$ is defined that may be different from $C_{1,t}$, and that also defined a second input as 0 or 1 for a second analyte. The response of such a system to the concentrations of both analytes in solution can be interpreted in the form of a logic table (Table 3). This approach has been proposed for diagnostic, as a diagnostic often does not consist in measuring quantitatively different concentrations of different analytes, but in assessing by a yes or no answer, and in an integrated fashion, if different parameters of a biological systems correspond to a healthy or a diseased state.

**Table 3 Logic table of an ideal endothelial dysfunction sensor.**

| $Ca^{2+}$ | Nitric oxide | Endothelial dysfunction |
|---|---|---|
| 0 | 0 | 0 |
| 1 | 0 | 1 |
| 0 | 1 | 0 |
| 1 | 1 | 0 |

For example, when comparing healthy and diseased endothelial cells, it appears that in healthy cells $Ca^{2+}$ is high (0.5–1 µM) and NO as well, while in diseased EC calcium is high, too (probably around 1 µM), but NO remains low. In the simplest approach, if sensors for $Ca^{2+}$ and NO would be available that are not toxic to endothelial cells, and would respond to biologically relevant concentrations of both analytes, then an ideal ED sensor would be a molecule (or a supramolecular system) that shows the logical table shown in Table 3. The response of the combined sensing platform should be 1, if $Ca^{2+}$ is high and NO low, and in all other cases, the response should be 0. This table is a simple functional table that does not consider *where* $Ca^{2+}$ and NO are detected. NO is membrane permeable, so the NO molecular probe may be either in blood, inside the EC or SMC, or at their surface. By contrast, $Ca^{2+}$ trafficking is strongly regulated as $Ca^{2+}$ ions are impermeable to biological membranes, so calcium molecular probes should be selectively taken up into the cytosol of endothelial cells. In addition, such a table corresponds to the simplest approach where the response of both sensors is essentially time-independent or provide a time-averaged response to $Ca^{2+}$ ions and NO. A more sophisticated sensing platform may give a response that combines time-dependent sensing of the concentration of both analytes. Ideally, such probes should targeted ECs, or at least be taken up preferentially (or more quickly) by EC. Overall, it appears that although a wide range of molecular probes for NO and calcium ions have been designed and prepared in the past, we are still far from an integrated diagnostic platform that would provide by a simple reading of a fluorescence signal, whether a given set of EC or SMC show signs of ED or not. Preparing such a sensing system remains a daunting challenge that will require active collaboration between synthetic chemists, photochemists, and biologists.

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

## Acknowledgements

This work has received funding from the European Union's Horizon 2020 research and innovation programme under the Marie Skłodowska-Curie grant agreement 813920 (LOGICLAB).

## Author contributions

V.D.A., R.C.A.K. and H.E. contributed equally in preparation of the manuscript. M.P. contributed to the writing and editing. S.B. and L.J.v.d.B. supervised the work and edited the manuscript.

## Competing interests

H.E. and L.J.v.d.B. are employees of Mimetas B.V., which markets the OrganoPlate and holds the registered trademarks OrganoPlate. The other authors declare no competing interests.

**Additional information**

