## [Peer Review File · Communications Chemistry]

Reviewers' comments:

Reviewer #1 (Remarks to the Author):

The review summarizes the background of endothelial dysfunction, and especially the role of calcium ions and NO in the detection of this clinical condition. Based on this fundament the currently available tools for the sensing of these analytes are presented and their pros and cons are discussed. Special emphasis is given to molecular fluorescent probes, including aspects such as sensitivity, selectivity, real-time measurements, and kinetic monitoring. In this reviewer's opinion the presented material reflects very well the state of the art of this specific topic. Not only from this viewpoint is the work highly recommended, but also because of the rather didactic/tutorial approach. For example, the authors present the basics of the implied photomechanistic phenomena (photoinduced electron transfer) and supramolecular binding events. They provide practical examples for the correct choice of probe concentrations and adequate ligand design for analyte binding.

I enjoyed reading this work and have only a few comments that may serve to improve the review further. With these minor revisions I would recommend acceptance of the work.

a) The use of NIR emission has been discussed as a preferable means to read the probes. Maybe a short comment could be included regarding the usually lower quantum yields of emissions in this spectral range, due to the low-lying emissive states and the resulting competitive non-radiative deactivation (cf., energy-gap law).

b) The sensing of NO in the absence of oxygen would be an advantage for probing under hypoxic conditions. Maybe this could be included somewhere.

c) The NO sensing depends on a chemical reaction between NO and o-phenylenediamine. Is there anything known about the timescale of this reaction?

d) The dependence of intensometric measurements on the probe concentration is discussed.

Ratiometric sensors are presented as a solution and this is of course totally valid. In this context, is there something known about lifetime-based probes, which in principle would enable monitoring independent on the amount of internalized probe?

e) For the NO sensors it would be useful to have a table with the spectral information such as excitation wavelength and detection (emission) wavelength, and, where available, some information about the dynamic range (fluorescence enhancement).

f) The use of amine electron donors as part of the receptor design bears usually interferences from protonation (pH-dependent) event. This should be briefly discussed.

Reviewer #2 (Remarks to the Author):

Summary: This review discusses the potential of molecular probes as sensors for nitric oxide and extracellular calcium as a means to quantify early stage endothelial dysfunction, which has significant implications in cardiovascular health. Overall, the review is well written (barring minor grammatical issues throughout) and extensive, covering a breadth of topics on this front. I appreciated the commitment to discussing the objectives and constraints of molecular probes designed for this purpose, and the focus on discussing technical considerations and limitations of nitric oxide and calcium sensors

will be useful to scientists in the field looking to design molecular probe sensors for these moieties. However, due to the breadth of topics, the focus of the review was lost at times. In addition, the review read primarily as a review of reviews rather than a review of recent advances in the primary literature. The review would benefit from an expanded discussion of recent advances in the primary literature, particularly on nitric oxide and calcium sensors for ED, which is the intended goal/focus of the review.

Comments and Questions:

- It is not clear to me how section 2 fits within the scope of the review. I suggest considering removal of section 2 to allow for expansion of sections 5 and 6, which I understand to be the intended focus of the review.
- In particular, sections 5.4, 5.5, and 5.6 would be strengthened by adding specific detail about recent primary literature (methods and results) that highlight the advances in molecular probes for measuring NO. For example, this review appears to be missing a discussion of recent advances in molecular probes conjugated to nanoparticles for increased signal, resolution, and localization. A deeper discussion of what is currently being done and their conclusions will also have the benefit of strengthening the discussion of the limitations, thereby demonstrating the necessity for the considerations that this review discusses.
- Section 6.3 mentions that only 'very few' molecular probes have been developed for calcium measurement for ED. The review would benefit from an in-depth discussion of these probes specifically and their limitations.
- I find myself a bit confused in section 5.2. Lines 338-344 appear to be highlighting some of the limitations of current molecular probes for NO detection, such as the requirement of electrodes or the need for extensive chemical manipulation. Line 345-364 then states that combining emission spectroscopy and optical microscopy is a better-suited solution. However, molecular probes are fluorescent dyes that use these methods to evaluate them, so I am unsure how the sentences relate. The section then continues in line 347-348 to say that fluorescence signal can be modulated by chemical modifications, which appears to be suggesting that this is a good thing, but a few lines above indicate it as a limitation.
- There are minor grammatical mistakes throughout that should be addressed.

Reviewer #1 (Remarks to the Author):

The review summarizes the background of endothelial dysfunction, and especially the role of calcium ions and NO in the detection of this clinical condition. Based on this fundament the currently available tools for the sensing of these analytes are presented and their pros and cons are discussed. Special emphasis is given to molecular fluorescent probes, including aspects such as sensitivity, selectivity, real-time measurements, and kinetic monitoring. In this reviewer's opinion the presented material reflects very well the state of the art of this specific topic. Not only from this viewpoint is the work highly recommended, but also because of the rather didactic/tutorial approach. For example, the authors present the basics of the implied photomechanistic phenomena (photoinduced electron transfer) and supramolecular binding events. They provide practical examples for the correct choice of probe concentrations and adequate ligand design for analyte binding.

I enjoyed reading this work and have only a few comments that may serve to improve the review further. With these minor revisions I would recommend acceptance of the work.

Response to the Reviewer:

We thank the reviewer for their appreciation of our review on nitric oxide and calcium sensors for endothelial dysfunction detection. We hope that our additions, answers, and modifications will be sufficient to alleviate any concerns.

- a) The use of NIR emission has been discussed as a preferable means to read the probes. Maybe a short comment could be included regarding the usually lower quantum yields of emissions in this spectral range, due to the low-lying emissive states and the resulting competitive non-radiative deactivation (cf., energy-gap law).
 - We agree. The limitation for NIR nitric oxide probes due to the energy gap law and newer strategies to overcome this limitation are mentioned in section 4.3. (page 7)
- b) The sensing of NO in the absence of oxygen would be an advantage for probing under hypoxic conditions. Maybe this could be included somewhere.
 - We agree. In section 5.4.3, a sentence was added to show the advantages of probing under hypoxic conditions (page 16).
- c) The NO sensing depends on a chemical reaction between NO and o-phenylenediamine. Is there anything known about the timescale of this reaction?
 - This is indeed an important question that is rarely discussed in the field. The question of the timescale of the reaction between NO and o-phenylenediamine is now included in section 5.4.2 of the revised version. The kinetics of this reaction involves an intermediate molecule, N_2O_3 , and the rate determining step is the formation of NO_2 from NO. (page 14)
- d) The dependence of intensometric measurements on the probe concentration is discussed. Ratiometric sensors are presented as a solution and this is of course totally valid. In this context, is there something known about lifetime-based probes, which in principle would enable monitoring independent on the amount of internalized probe?
 - Good question indeed. A time-resolved technique which is concentration independent is time-resolved emission spectroscopy. Lifetime of the excited species might change in

case of dynamic “quenching”, or interaction, of analyte with the sensor. Small molecule sensors of NO and Ca²⁺ are, by definition, based on static interaction of the analyte and the sensor in their ground states, thus the sensor emission lifetime stays the same and could not be used as a changing output of the sensor. At the same time the lifetime-based sensors might be very interesting for analytes interacting with sensors in their excited states. A note has been added in the end of part 6.2.2.1.

- e) For the NO sensors it would be useful to have a table with the spectral information such as excitation wavelength and detection (emission) wavelength, and, where available, some information about the dynamic range (fluorescence enhancement).
- Agreed. A new table of nitric oxide sensors with excitation wavelength, emission wavelength, selectivity details, limit of detection, tested cell details and kinetics of fluorescence was added as Table 1a (page 29). Table 1b includes a selection of molecular probes for Ca²⁺.
- f) The use of amine electron donors as part of the receptor design bears usually interferences from protonation (pH-dependent) event. This should be briefly discussed.
- We agree. A paragraph about the pH dependence of amine-based nitric oxide sensors was included in section 5.4.2. Potential methods that can be adopted to overcome this limitation were also mentioned. (page 14)

Reviewer #2 (Remarks to the Author):

Summary: This review discusses the potential of molecular probes as sensors for nitric oxide and extracellular calcium as a means to quantify early stage endothelial dysfunction, which has significant implications in cardiovascular health. Overall, the review is well written (barring minor grammatical issues throughout) and extensive, covering a breadth of topics on this front. I appreciated the commitment to discussing the objectives and constraints of molecular probes designed for this purpose, and the focus on discussing technical considerations and limitations of nitric oxide and calcium sensors will be useful to scientists in the field looking to design molecular probe sensors for these moieties. However, due to the breadth of topics, the focus of the review was lost at times. In addition, the review read primarily as a review of reviews rather than a review of recent advances in the primary literature. The review would benefit from an expanded discussion of recent advances in the primary literature, particularly on nitric oxide and calcium sensors for ED, which is the intended goal/focus of the review.

Response to Review 2:

- We thank the reviewer for their feedback. We hope that our additions, answers, and modifications will be sufficient to alleviate any concerns.

- It is not clear to me how section 2 fits within the scope of the review. I suggest considering removal of section 2 to allow for expansion of sections 5 and 6, which I understand to be the intended focus of the review.
 - We respectfully disagree. We assume the referee to be an expert in the field of ED and its modelling, but our teams includes both biologists and pure chemists for whom the biological information contained in part 2 was very new. We assume it can also be the case for the future readers of this article. Section 2 helps put the disease in context and helps to identify why NO and calcium are important molecules to detect. We agree, however, that we could focus this part. Section 2 and 3 were hence combined and given a more descriptive title to help emphasize the importance of NO and calcium in ED (page 2).
- In particular, sections 5.4, 5.5, and 5.6 would be strengthened by adding specific detail about recent primary literature (methods and results) that highlight the advances in molecular probes for measuring NO. For example, this review appears to be missing a discussion of recent advances in molecular probes conjugated to nanoparticles for increased signal, resolution, and localization. A deeper discussion of what is currently being done and their conclusions will also have the benefit of strengthening the discussion of the limitations, thereby demonstrating the necessity for the considerations that this review discusses.
 - We agree and thank the referee for noticing this missing examples. A completely new section 5.4.4 (page 16) has been added to include these recent nanoparticle-based nitric oxide sensors.
- Section 6.3 mentions that only ‘very few’ molecular probes have been developed for calcium measurement for ED. The review would benefit from an in-depth discussion of these probes specifically and their limitations.
 - Agree. Section 6.3 has been corrected to include the most promising calcium probes for ED imaging (Table 1b on page 30).
- I find myself a bit confused in section 5.2. Lines 338-344 appear to be highlighting some of the limitations of current molecular probes for NO detection, such as the requirement of electrodes or the need for extensive chemical manipulation. Line 345-364 then states that combining emission spectroscopy and optical microscopy is a better-suited solution. However, molecular probes are fluorescent dyes that use these methods to evaluate them, so I am unsure how the sentences relate. The section then continues in line 347-348 to say that fluorescence signal can be modulated by chemical modifications, which appears to be suggesting that this is a good thing, but a few lines above indicate it as a limitation.
 - Point taken. Some of the sentences have been rephrased to convey the information in of section 5.2 in a clearer manner (page 10).
- There are minor grammatical mistakes throughout that should be addressed.
 - Thank you for the advice. We had the grammar checked.

REVIEWERS' COMMENTS:

Reviewer #1 (Remarks to the Author):

The authors present a revised version of their manuscript. They have included new discussion and additional data, which have improved the review further. All of my comments have been addressed adequately. As far as I can see also the comments of the other reviewer were dealt with in an appropriate manner. The manuscript has gained further quality and represents a very informative and didactic review of the topic. I recommend the acceptance of the work in its present state.

Reviewer #2 (Remarks to the Author):

I am satisfied with the authors' response and edits.